# The Stat3-Fam3a axis promotes muscle stem cell myogenic lineage progression by inducing mitochondrial respiration

David Sala[1], Thomas J. Cunningham[1,4], Michael J. Stec[1], Usue Etxaniz [1], Chiara Nicoletti [1], Alessandra Dall'Agnese[1], Pier Lorenzo Puri [1,2], Gregg Duester [1], Lucia Latella[2,3] & Alessandra Sacco[1]

Metabolic reprogramming is an active regulator of stem cell fate choices, and successful stem cell differentiation in different compartments requires the induction of oxidative phosphorylation. However, the mechanisms that promote mitochondrial respiration during stem cell differentiation are poorly understood. Here we demonstrate that Stat3 promotes muscle stem cell myogenic lineage progression by stimulating mitochondrial respiration in mice. We identify Fam3a, a cytokine-like protein, as a major Stat3 downstream effector in muscle stem cells. We demonstrate that Fam3a is required for muscle stem cell commitment and skeletal muscle development. We show that myogenic cells secrete Fam3a, and exposure of Stat3-ablated muscle stem cells to recombinant Fam3a in vitro and in vivo rescues their defects in mitochondrial respiration and myogenic commitment. Together, these findings indicate that Fam3a is a Stat3-regulated secreted factor that promotes muscle stem cell oxidative metabolism and differentiation, and suggests that Fam3a is a potential tool to modulate cell fate choices.

[1] Development, Aging and Regeneration Program, Sanford Burnham Prebys Medical Discovery Institute, 10901N Torrey Pines Road, La Jolla, CA 92037, USA. [2] IRCCS, Fondazione Santa Lucia, Rome 00142, Italy. [3] Institute of Translational Pharmacology, National Research Council of Italy, Via Fosso del Cavaliere 100, Rome 00133, Italy. [4] Present address: Mammalian Genetics Unit, MRC Harwell Institute, Oxfordshire OX11 0RD, UK. Correspondence and requests for materials should be addressed to A.S. (email: asacco@sbpdiscovery.org)

Accumulating evidence indicates that metabolism is not only a consequence of the stem cell functional status but rather an active player that regulates stem cell fate choices[1]. Proliferative stem cells rely on glycolysis to obtain the energy they require, as well as to generate the necessary metabolic intermediates to sustain their growth[2,3]. Next, the switch from self-renewal towards commitment and differentiation requires the induction of oxidative metabolism[2,3]. Stimulation of mitochondrial respiration during cell differentiation is a common feature shared by a wide range of stem cell compartments, including hematopoietic stem cells[4,5], neuronal stem cells[6,7], mesenchymal stem cells[1,8], embryonic stem cells[1], and muscle stem cells (MuSCs)[2,9–12]. However, the mechanisms that induce mitochondrial function for successful stem cell differentiation are still poorly understood.

MuSCs are responsible for skeletal muscle formation during development, as well as for the maintenance of tissue homeostasis and repair during adulthood[13]. MuSC transition between different functional stem cell states can be identified by the expression of myogenic markers, which are transcription factors dynamically expressed during MuSC myogenic lineage progression[13]. This makes MuSCs a powerful tool for studying the mechanisms that regulate stem cell fate choices. MuSCs reside in a quiescent state in adult skeletal muscle. Upon injury, MuSCs become activated, proliferate, and differentiate to form new myofibers and repair the tissue[14]. Skeletal muscle regeneration is a very efficient process, and temporally coordinated changes in metabolism play a central role in regulating proper MuSC myogenic lineage progression to ensure successful muscle regeneration[2,9–12,15]. Quiescent MuSCs display a low metabolic rate that mainly relies on fatty acid oxidation[2,15]. MuSC activation initially induces their glycolytic pathway in order to sustain cell proliferation[2,15], and then a switch to oxidative metabolism is required to allow further MuSC commitment to the myogenic lineage and differentiation[2,9–12]. Several pathological conditions and aging can compromise MuSC function, alter the balance between self-renewal and differentiation, and result in the reduction of skeletal muscle regenerative capacity that contributes to the muscle loss associated with these diseases[16–18]. Thus identifying the factors that regulate MuSC metabolism and therefore cell fate choices is highly relevant for therapeutic purposes.

We and others recently provided evidence that signal transducer and activator of transcription factor 3 (Stat3) regulates MuSC fate choices during skeletal muscle regeneration[19,20]. Stat3 is a transcription factor that mediates the intracellular signaling of several cytokines, including interleukin (IL)-6, leukemia inhibitory factor, and Oncostatin M[21]. We demonstrated that Stat3 promotes MuSC progression into committed myogenic progenitors and differentiation[20]. However, the mechanism by which Stat3 regulates MuSC myogenic commitment is currently poorly understood. Stat3 regulates a wide range of biological processes in different cell types, including proliferation, migration, survival, and metabolism[21]. From the metabolic point of view, Stat3 is able to promote glycolysis or mitochondrial respiration depending on the cellular context[22–24]. However, little is currently known about the role of Stat3 in regulating MuSC metabolism.

In our study, we demonstrate that Stat3 promotes mitochondrial respiration during MuSC myogenic lineage progression. We further identify Fam3a as a regulator of MuSC function that acts as a major direct downstream effector of Stat3. Fam3a is a cytokine-like protein shown to increase ATP production and promote mitochondrial respiration in vascular smooth muscle cells, neuronal cells, and hepatocytes[25–27]. We demonstrate that Fam3a is required for proper MuSC myogenic lineage progression and skeletal muscle development in vivo. We further show that Fam3a is secreted by myogenic cells and that treatment with recombinant Fam3a rescues the reduced mitochondrial respiration and defective myogenic commitment of Stat3-ablated MuSCs both in vitro and in vivo during adult skeletal muscle repair. Overall, this work positions the Stat3–Fam3a axis as a driver of mitochondrial respiration during MuSC commitment and differentiation and suggests that therapeutic interventions targeting this axis could be utilized to promote MuSC-mediated tissue repair.

## Results

**Stat3 promotes mitochondrial function in MuSCs.** In order to analyze the impact of Stat3 on MuSC gene expression during activation, we performed RNA-seq whole-transcriptome analysis in control and Stat3 genetically ablated MuSCs (Stat3 knockout (KO)). To this aim, we took advantage of the Pax7-CreER driver, as Pax7 is a transcription factor specifically expressed in MuSCs in skeletal muscle[13,20]. We induced Cre-mediated Stat3 ablation in 3-month-old Pax7-CreER;Stat3[f/f] male mice and control littermates (Pax7-CreER[WT];Stat3[f/f]) by tamoxifen (Tmx) treatment, as previously reported (Fig. 1a)[20]. At least 2 weeks after treatment, tibialis anterior muscles were injured by intramuscular injection of barium chloride (BaCl$_2$) to induce MuSC activation. MuSCs were isolated from uninjured or 3 days post-injury (3 dpi) mice by fluorescence-activated cell sorting (FACS), based on α7-integrin and CD34 cell surface marker expression, as previously reported (Fig. 1a)[20,28]. Comparison between control uninjured and control activated MuSCs (3 dpi) showed major changes in their transcriptome (Fig. 1b). Pathway analysis (using Gene Set Enrichment Analysis (GSEA)[29]) revealed that the top enriched pathways during activation of control (Ct) MuSCs were related to cell cycle, protein synthesis, and mitochondrial metabolism, consistent with previous reports (Fig. 1c, Supplementary Fig. 1a, and Supplementary Data 1)[9]. Activation of Stat3 KO MuSCs also caused extensive changes in their transcriptome (activated—3 dpi —vs uninjured Stat3 KO MuSCs) (Fig. 1b). GSEA showed that pathways associated with cell cycle and protein synthesis were among the top 20 enriched pathways in activated Stat3 KO MuSCs compared to uninjured Stat3 KO MuSCs (Fig. 1c, Supplementary Fig. 1b, and Supplementary Data 2). In contrast to what we observed during activation of control MuSCs, none of the top 20 enriched pathways in activated Stat3 KO MuSCs were associated with mitochondrial oxidative phosphorylation (Fig. 1c, Supplementary Fig. 1b), suggesting that Stat3 may play a central role in inducing mitochondrial function during MuSC activation.

We further interrogated the data by performing pathway analysis using the differentially expressed transcripts between activated control and activated Stat3 KO MuSCs. GSEA identified previously described Stat3-regulated pathways such as integrin signaling, cytokine–cytokine receptor interaction, and Janus-activated kinase–Stat signaling among the top enriched pathways (Fig. 1d and Supplementary Data 3). The respiratory electron transport pathway appeared in the top 10 differentially enriched pathways, further suggesting that Stat3 promotes mitochondrial function during the activation process (Fig. 1d and Supplementary Data 3). Thus we analyzed the mitochondrial respiration of MuSCs lacking Stat3 after 3 days in culture in growth conditions. Consistent with our pathway analysis, Stat3-ablated MuSCs exhibited a decreased mitochondrial respiration assessed as reduced basal and maximal oxygen consumption rate (OCR; Fig. 1e, f). Stat3 ablation did not affect the extracellular acidification rate (ECAR), readout for glycolysis (Fig. 1g). Overall, this data indicates that Stat3 promotes mitochondrial respiration during MuSC commitment to the myogenic lineage without having a major impact on glycolysis.

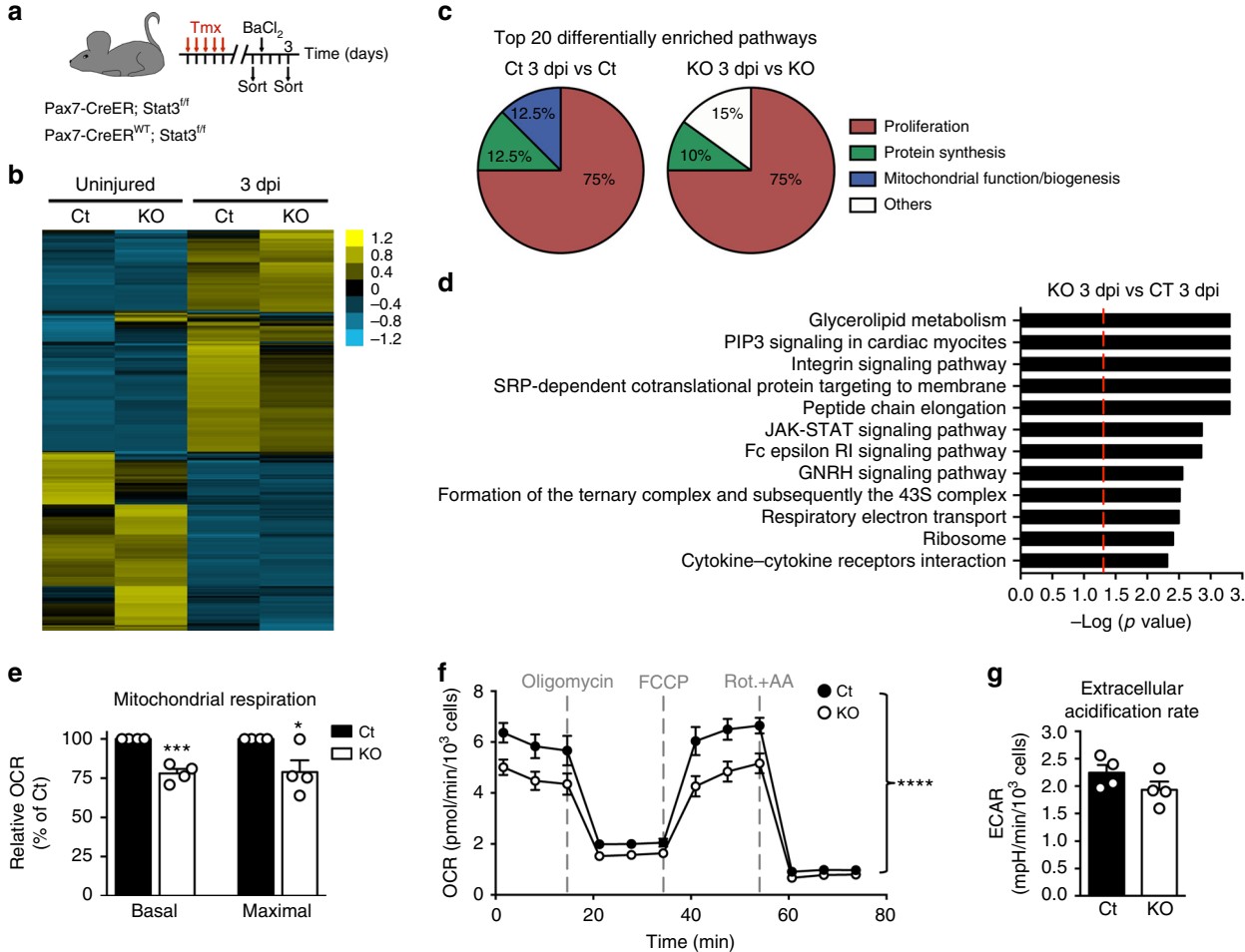

**Fig. 1** Stat3 promotes mitochondrial function in muscle stem cells (MuSCs) during activation. **a** Scheme of the experimental design to obtain the samples for RNA-seq. **b** Heat map of the differentially expressed genes among the different comparisons. Listed genes were differentially expressed in at least one of the comparisons. Normalized gene RPKM (reads per kilobase of transcript per million) values were averaged within groups for the generation of the heat map (n = 3 animals). **c** Pie charts showing the top 20 differentially enriched pathways (using Gene Set Enrichment Analysis (GSEA)) during the activation of control (Pax7-CreER[WT];Stat3[f/f]) or Stat3 KO MuSCs (Pax7-CreER;Stat3[f/f] mice) grouped according to their functional category. **d** Top 12 pathways enriched in freshly isolated MuSCs from 3 dpi Pax7-CreER;Stat3[f/f] mice compared to 3 dpi MuSCs isolated from control littermates using GSEA. The red line indicates p value = 0.05 based on the pathway analysis (GSEA). **e–g** Measurement of the oxygen consumption rate (OCR) and the extracellular acidification rate (ECAR) of control and Stat3 KO MuSCs cultured in growth conditions for 3 days (n = 4 independent experiments). The Cell Mito Stress Test was performed in a Seahorse XFp Analyzer. Data are represented as mean ± SEM (Student's t test or two-way analysis of variance; *p < 0.05, ***p < 0.001, ****p < 0.0001)

Stat3 has been shown to promote mitochondrial respiration by translocating into the mitochondria[22–24]. In order to assess a potential role of mitochondrial Stat3 during MuSC activation, we performed immunofluorescence analysis of both total and S727 phosphorylated Stat3 in MuSCs cultured in vitro for 3 days. Stat3 S727 phosphorylation enhances its transcriptional activity in the nucleus, and it is also required to induce the translocation of Stat3 into the mitochondria[24,30,31]. Both analyses showed a predominant nuclear localization for Stat3 (Supplementary Fig. 1c, d), suggesting that nuclear Stat3 plays a major role in regulating mitochondrial respiration during MuSC activation.

Previous studies demonstrated that mitochondrial respiration actively promotes myogenic differentiation in cultured myoblasts[32–36]. Consistently, mitochondrial biogenesis and mitochondrial respiration are increased during MuSC myogenic commitment and differentiation[2,9–11,37]. Thus, to further validate that mitochondrial respiration is an important factor that promotes myogenic progression in MuSCs, we cultured MuSCs in vitro in growth media (GM) or differentiation media (DM) and

treated them with carbonyl cyanide 3-chlorophenylhydrazone (CCCP) (or dimethyl sulfoxide, DMSO, as vehicle), an inhibitor of oxidative phosphorylation, for 24 h. In both contexts, the percentage of myogenin-positive cells was reduced in CCCP-treated MuSCs (Supplementary Fig. 1e, f), indicating that mitochondrial respiration is required for proper myogenic commitment and differentiation in MuSCs.

Overall, our findings indicate that Stat3-dependent regulation of mitochondrial respiration is a major mechanism that promotes MuSC commitment and differentiation.

**Fam3a is a downstream target of Stat3 and MyoD in MuSCs.** To identify relevant downstream targets of Stat3 that promote MuSC mitochondrial respiration and myogenic lineage progression, we performed multiple comparisons between all the transcriptomes obtained from the RNA-seq analysis (Fig. 2a). We focused our attention on genes that were differentially expressed during the process of activation in control MuSCs (Ct 3 dpi vs Ct),

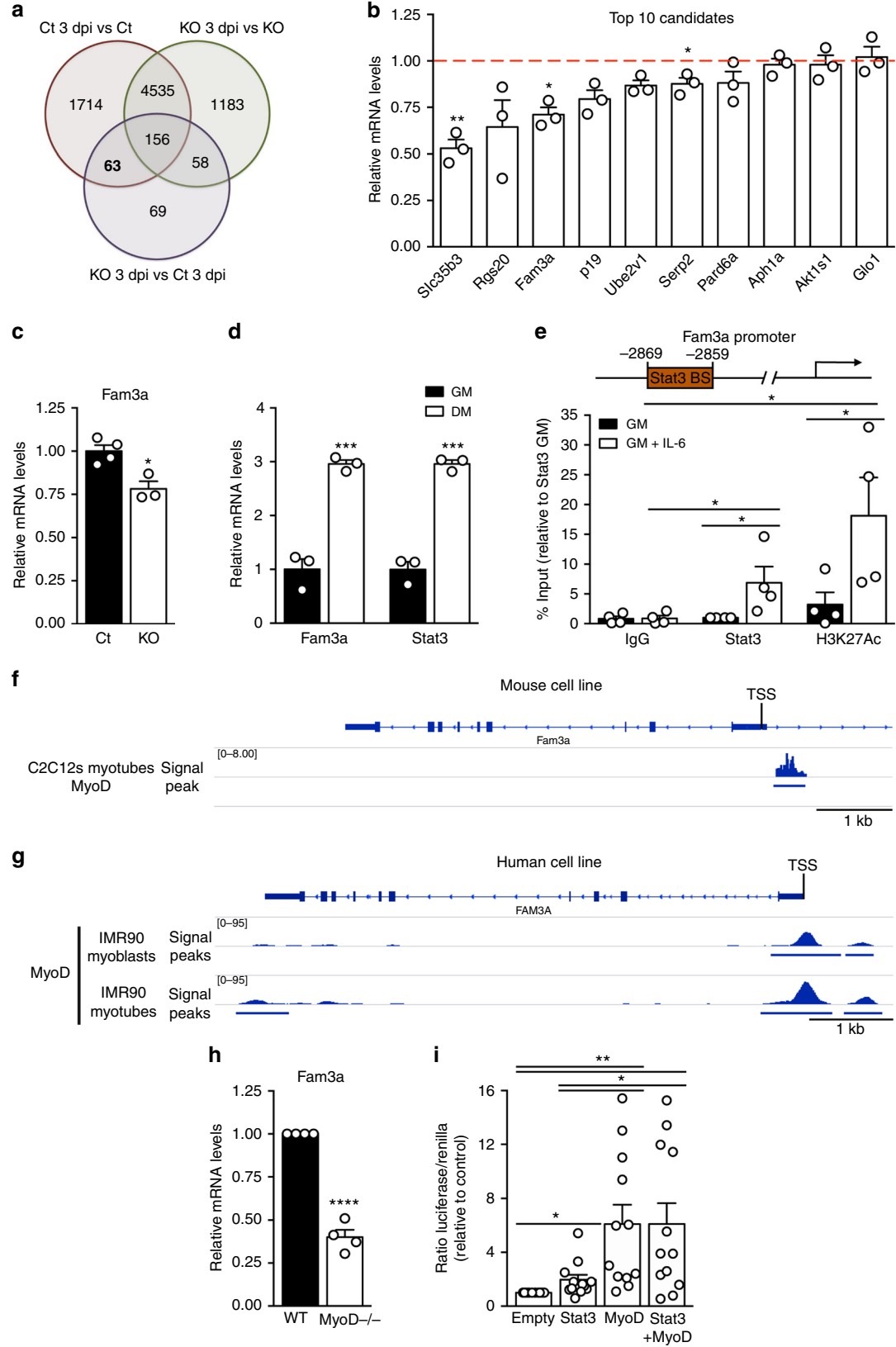

that did not respond during the activation of Stat3 KO MuSCs (KO 3 dpi vs KO), and that were differentially expressed between activated Stat3 KO MuSCs and activated control MuSCs (KO 3 dpi vs Ct 3 dpi). This analysis identified 63 genes (Fig. 2a). Stat3 is

primarily a transcriptional coactivator[38], thus we concentrated on the top 10 genes downregulated in activated Stat3 KO MuSCs (KO 3 dpi) compared to activated control cells (Ct 3 dpi). Among the significantly downregulated genes (Fig. 2b), we focused on

**Fig. 2** Fam3a is a direct Stat3 and MyoD target. **a** Venn diagram of the overlap among the differentially expressed genes between the different comparisons in the RNA-seq analysis. **b** Quantitative real-time PCR analysis of the top 10 selected candidates in the RNA-seq samples ($n = 3$ animals). The red line indicates the average expression levels in control samples. **c** Fam3a mRNA levels in freshly isolated activated muscle stem cells (MuSCs) (3 dpi) from Pax7-CreER;Stat3[f/f] mice and control littermates (different samples from RNA-seq) ($n = 3$–4 animals). **d** Gene expression analysis of MuSCs cultured in growth media (GM) or differentiation media (DM) for 72 h ($n = 3$ independent experiments). **e** Fam3a promoter analysis using JASPAR identified one putative Stat3-binding site. Chromatin immunoprecipitation (ChIP) assay in C2C12 myoblasts incubated in the presence or absence of interleukin-6 (100 ng/ml) to monitor the recruitment of Stat3 and the presence of histone H3 acetylated at lysine 27 (H3K27Ac) in the Fam3a promoter ($n = 4$ independent experiments). **f** MyoD ChIP-seq signal and peak distribution on the Fam3a promoter on C2C12 myotubes. Previously published data were used for the analysis[40]. **g** MyoD ChIP-seq signal distribution and peaks on the FAM3A promoter on IMR90-derived myoblasts and myotubes. **h** Fam3a mRNA levels in wild type and MyoD−/− MuSCs cultured in growth conditions for 72 h ($n = 4$ independent experiments). **i** Reporter assay in HEK293 cells transfected with plasmids encoding Stat3, MyoD, and/or an empty vector as control, together with the reporter vector containing the luciferase reporter gene under the control of the Fam3a promoter ($n = 12$ independent experiments). Data are represented as mean ± SEM (Student's $t$ test; *$p < 0.05$, **$p < 0.01$, ***$p < 0.001$, ****$p < 0.0001$)

---

Fam3a, a cytokine-like protein, as it was the only one that had been reported to be involved in the regulation of mitochondrial metabolism[25–27].

We validated *Fam3a* downregulation in activated Stat3 KO MuSCs compared to activated controls in samples different from the RNA-seq (Fig. 2c). We further observed upregulation of *Fam3a* at the mRNA level in MuSCs during myogenic differentiation in vitro, mirroring the expression pattern of Stat3 (Fig. 2d). To investigate whether *Fam3a* is a direct transcriptional target of Stat3, we performed *Fam3a* promoter analysis using JASPAR[39] and identified one putative Stat3-binding site 2869 bp upstream of the transcription start site (TSS; Fig. 2e). Chromatin immunoprecipitation (ChIP) assay in C2C12 myoblasts showed that Stat3 is recruited to this site upon IL-6 stimulation, which promotes Stat3 activation and translocation into the nucleus (Fig. 2e). IL-6 treatment also caused enrichment of H3K27Ac, a marker of active transcription, in this region (Fig. 2e). Together, these findings indicate that *Fam3a* is a direct transcriptional target of Stat3.

Further analysis of the *Fam3a* promoter revealed the existence of putative MyoD-binding sites. MyoD is a transcription factor essential for MuSC commitment to the myogenic lineage and differentiation[13], and recent work demonstrated that MyoD regulates a set of genes responsible to sustain oxidative metabolism in C2C12 myotubes and adult skeletal muscle[10]. By analyzing previously published ChIP-seq data[40], we observed MyoD binding to the *Fam3a* promoter in proximity to the TSS in C2C12 myotubes (Fig. 2f). Similarly, ChIP-seq analysis using myogenic conversion of human IMR90 fibroblasts to the myogenic lineage by the induction of ectopic MyoD expression showed the recruitment of MyoD to the *FAM3A* promoter (Fig. 2g). This MyoD recruitment was further increased by the induction of differentiation in myogenically converted IMR90 fibroblasts (Fig. 2g), suggesting that MyoD regulation of *Fam3a* is conserved between mouse and human species. Consistent with ChIP-seq data, MuSCs isolated from MyoD KO mice[41,42] showed reduced *Fam3a* mRNA levels when cultured for 3 days in vitro (Fig. 2h).

Finally, to further validate that Stat3 and MyoD regulate *Fam3a* expression, we performed reporter assays using a construct containing the luciferase reporter gene under the control of the *Fam3a* promoter. HEK293 cells were transiently transfected with the reporter plasmid and a Renilla encoding plasmid (to monitor transfection efficiency), together with plasmids encoding for Stat3 and/or MyoD (Fig. 2i). Stat3 overexpression significantly increased the transcriptional activity of the reporter compared to control conditions, and MyoD overexpression induced a much higher transcriptional activation of the reporter (Fig. 2i). However, we did not observe an additive effect when transfecting together Stat3 and MyoD coding plasmids (Fig. 2i).

Altogether, our data indicate that both Stat3 and MyoD directly promote *Fam3a* expression by binding to the respective regulatory regions on the *Fam3a* promoter.

**Fam3a promotes MuSC myogenic lineage progression in vitro.** To analyze the function of Fam3a in MuSCs, we performed acute loss-of-function studies in vitro by utilizing lentiviruses expressing short hairpin RNA (shRNA) against Fam3a (shFam3a) or shRNA control (shCt). Freshly isolated MuSCs from 3-month-old C57BL6/J male mice were placed in culture in growth conditions, infected with lentiviruses, and after 72 h we analyzed myogenic lineage progression. We assessed the expression levels of the myogenic regulators Pax7 (marker of quiescent and proliferating MuSCs), MyoD (early marker of committed progenitors), and myogenin (late marker of committed progenitors). *Fam3a* mRNA levels were efficiently reduced with shRNA treatment, and this resulted in a reduction in myogenin mRNA (Fig. 3a). Immuno-fluorescence analysis showed that Fam3a knockdown increased the percentage of Pax7+MyoD− and Pax7+MyoD+ populations and reduced the percentage of the Pax7−MyoD+ population (Fig. 3b). We also observed a reduction in the percentage of myogenin+ cells, consistent with our mRNA data, indicating a reduced rate of commitment into the myogenic lineage upon Fam3a repression (Fig. 3c). No differences in proliferation were observed (Supplementary Fig. 2), suggesting that Fam3a specifically affects MuSC commitment. Upon induction of terminal differentiation, MuSCs knocked down for Fam3a exhibited impaired ability to differentiate into myosin heavy chain-positive (MyHC+) cells, as shown by a reduced differentiation index (Fig. 3d). Together, these findings demonstrate that Fam3a is required for proper myogenic lineage progression in vitro and its repression recapitulates the phenotype previously described for Stat3 gene deletion/inhibition in MuSCs[20], suggesting that Fam3a is a relevant downstream effector of Stat3 in MuSCs.

**Fam3a is required for skeletal muscle development in vivo.** To investigate the role of Fam3a in skeletal muscle in vivo, we generated a Fam3a KO mouse model by utilizing CRISPR/Cas9 genome-editing strategies (Fig. 4a). We designed a guide to specifically target exon 2 and generate frame-shift mutations that disrupt the open reading frame (Supplementary Fig. 3a). Fam3a contains an N-terminal secretion signal peptide (SP) followed by the sequence corresponding to the mature protein. With this strategy, we generated frame-shift mutations in the region encoding for the SP and therefore disrupted the protein before any part of the mature and active Fam3a protein is produced (Supplementary Fig. 3a).

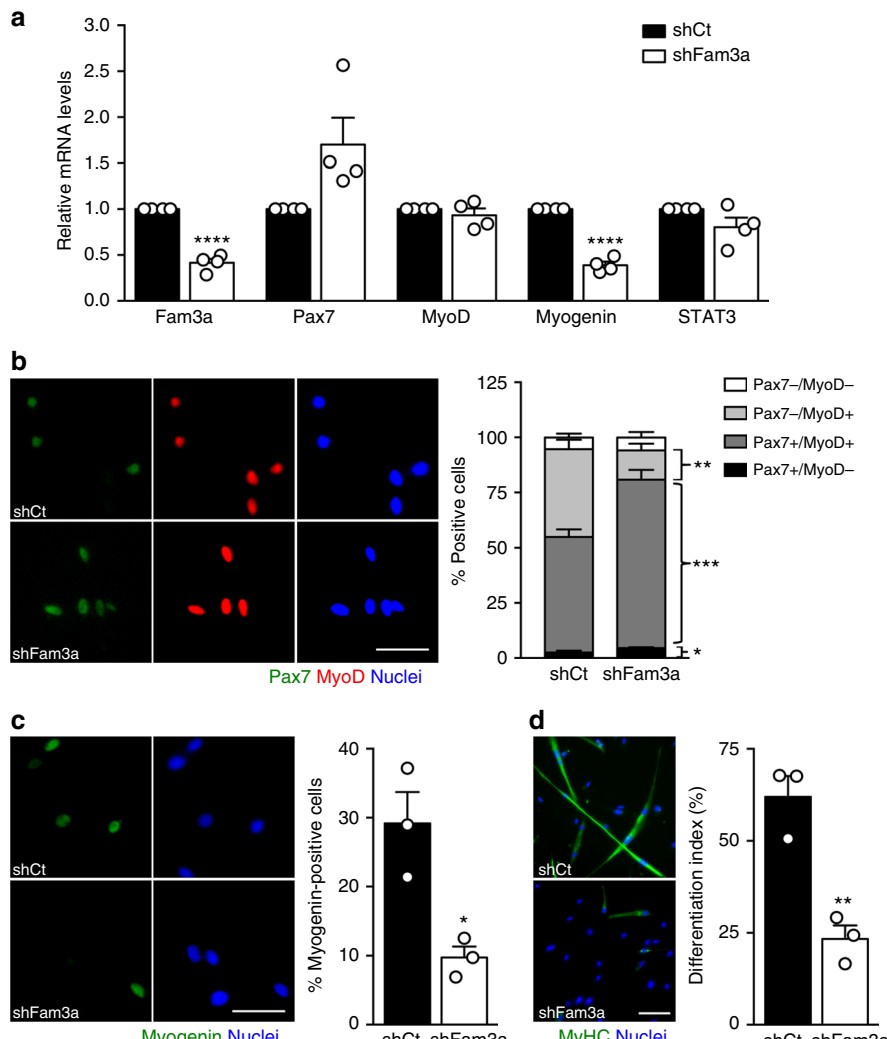

Fig. 3 Fam3a promotes muscle stem cell (MuSC) myogenic lineage progression in vitro. **a** Gene expression analysis of MuSCs infected with lentiviruses coding for short hairpin RNA (shRNA) Control (shCt) or shRNA against Fam3a (shFam3a) and cultured in growth conditions for 72 h ($n = 4$ independent experiments). **b**, **c** Immunofluorescence analysis and quantification of myogenic markers of MuSCs infected with shCt or shFam3a coding lentiviruses and cultured in growth conditions for 72 h ($n = 3$ independent experiments). Scale bars 50 µm. **d** Immunofluorescence analysis of myogenic differentiation of MuSCs infected with shCt or shFam3a coding lentiviruses and cultured in differentiation conditions for 72 h ($n = 3$ independent experiments). Scale bar 100 µm. Data are represented as mean ± SEM (Student's $t$ test; *$p < 0.05$, **$p < 0.01$, ***$p < 0.001$, ****$p < 0.0001$)

We injected the guide and Cas9 mRNAs into zygotes and assessed skeletal muscle development in E15 embryos and P0 pups (Fig. 4a). As Fam3a is an X chromosome-encoded gene, we focused our studies on males. DNA sequencing of the 5 Fam3a KO embryos analyzed in this study showed a frame-shift mutation in exon 2 (Supplementary Fig. 3b). Fam3a KO embryos exhibited significantly smaller hind limb muscles, shown by the reduced embryonic myosin heavy chain (eMyHC) area (Fig. 4b), as well as a reduction in the number of myogenin+ nuclei (Fig. 4c, d), compared to controls. We further analyzed Fam3a KO male mice at birth (P0). The three Fam3a KO pups used in this study displayed a frame-shift mutation in exon 2 (Supplementary Fig. 3b). Also in this context, we observed a reduction in eMyHC area and an accumulation of uncommitted MuSCs (assessed as increased number of Pax7+ cells) (Fig. 4e, f). Additionally, laminin was also more disrupted in Fam3a KO pups compared to controls (Fig. 4e, f). These findings are consistent with a role of Fam3a in promoting myogenic commitment and muscle differentiation during development, with its ablation leading to

a deficient MuSC progression into the myogenic lineage in vivo and compromised skeletal muscle formation.

To assess whether the reduction in muscle mass was maintained in adulthood, we analyzed body and tissue weights from 3-month-old wild-type (WT) and Fam3a KO male mice. Indeed, Fam3a deletion resulted in decreased body weight, together with a significant reduction in the gastrocnemius, quadriceps, and heart weights (Supplementary Fig. 4). We did not observe differences in the weight of other tissues such as liver, epididymal white adipose tissue, spleen, or kidneys (Supplementary Fig. 4). Together, this data indicates that Fam3a ablation causes a reduction in skeletal muscle mass during development that is maintained in adulthood.

Fam3a is known to induce mitochondrial respiration in several cell types[25–27]. Thus we assessed whether Fam3a also promotes mitochondrial function in MuSCs. To this aim, we measured the mitochondrial respiration in WT and Fam3a KO MuSCs cultured in growth conditions for 3 days. Fam3a genetic deletion reduced both basal and maximal mitochondrial respiration in MuSCs

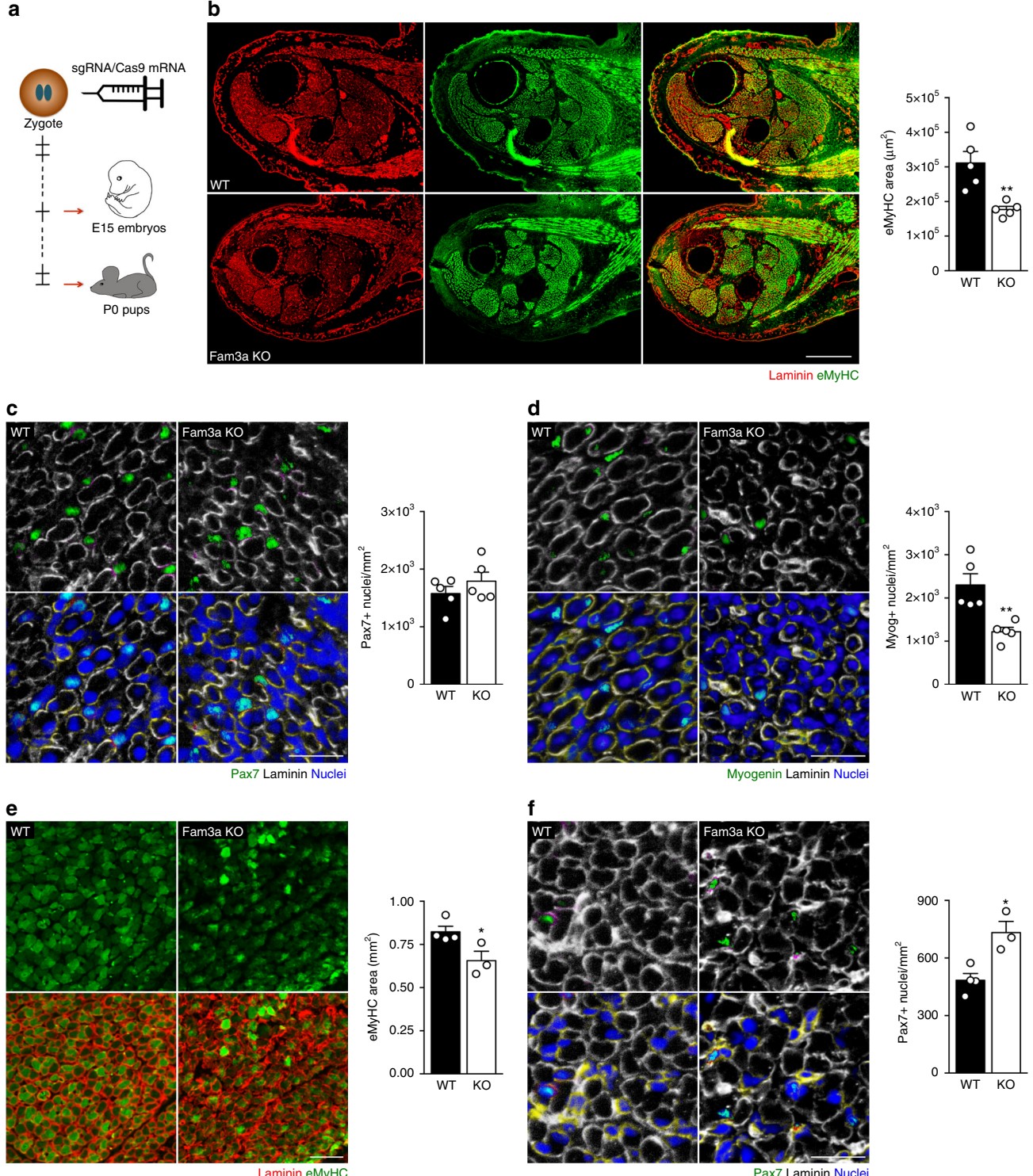

**Fig. 4** Fam3a ablation compromises skeletal muscle development in vivo. **a** Scheme of the generation of the Fam3a knockout (KO) mouse model and the experimental design. **b–d** Immunofluorescence analysis of transversal sections of the hind limbs of wild-type (WT) and Fam3a KO E15 embryos. Quantification of the embryonic myosin heavy chain (eMyHC) area and the number of Pax7- and Myogenin-positive cells ($n = 5$ embryos). Scale bars 300 and 30 μm. **e, f** Immunofluorescence analysis of transversal sections of the hind limbs of WT and Fam3a KO P0 pups. Quantification of the eMyHC area and the number of Pax7-positive cells ($n = 3$–4 pups). Scale bars 100 and 25 μm. Data are represented as mean ± SEM (Student's $t$ test; *$p < 0.05$, **$p < 0.01$)

without affecting glycolysis (Supplementary Fig. 5a–c). This reduction in mitochondrial oxygen consumption was not due to reduced mitochondrial content or changes in the expression of Pgc1a or different subunits of the electron transport chain in MuSCs (Supplementary Fig. 5d–f). Consistent with this data, Fam3a overexpression in C2C12 myoblasts did not affect the expression of Pgc1a or different subunits of the electron transport chain at the mRNA or protein level (Supplementary Fig. 5g–j). Together, this data demonstrates that Fam3a does not promote mitochondrial respiration by regulating mitochondrial content, biogenesis, or the expression of the electron transport chain complexes in myogenic cells, suggesting that alternative molecular mechanisms are taking place.

Overall, our data indicate that Fam3a is required for proper skeletal muscle development by stimulating mitochondrial respiration in MuSCs and promoting their myogenic lineage progression.

**Fam3a is secreted by myogenic cells.** Fam3a protein sequence analysis using LocTree3[43] predicted that Fam3a is a secreted protein (Supplementary Fig. 6a). However, previous reports indicated a localization of Fam3a in the mitochondria[25–27]. To strengthen our prediction, we used two additional softwares that predict protein localization: TargetP[44] and MitoFates[45]. TargetP is a software that predicts protein localization by analyzing the presence of N-terminal presequences containing mitochondrial targeting peptides or secretory SPs[44]. TargetP analysis of the Fam3a protein sequence also predicted that Fam3a is a secreted protein, while it correctly predicted that the protein Citrate Synthase was mitochondrial (Supplementary Fig. 6a). Finally, we used MitoFates as software that analyzes the presence of N-terminal mitochondrial targeting signals and their cleavage sites[45]. Consistent with the previous predictions, Fam3a did not contain any mitochondrial localization presequence while it was identified in Citrate Synthase (Supplementary Fig. 6b). Overall, the analysis of the Fam3a protein sequence using three different softwares indicated that Fam3a is a secreted protein that lacks a mitochondrial localization signal.

To directly investigate the subcellular localization of Fam3a in muscle cells, we performed colocalization studies in transiently transfected C2C12 myoblasts with a construct expressing Fam3a-Myc-Flag. In order to label Golgi, endoplasmic reticulum (ER), and mitochondria, we used the markers GM-130, KDEL, and Tomm20, respectively. In accordance with previously published studies, these markers were specific for each compartment as they presented no or minimal colocalization among them (Supplementary Fig. 6c, d)[46–49]. Consistent with our Fam3a localization predictions, we did not detect colocalization between exogenous Fam3a and mitochondria 48 h after transfection (Supplementary Fig. 6d, e). Instead, exogenous Fam3a colocalized with GM-130, marker of cis-Golgi (Fig. 5a and Supplementary Fig. 6d). Upon treatment with the secretion inhibitor monensin, Fam3a-Myc-Flag strongly colocalized with the ER, as shown by colocalization with KDEL (Fig. 5b and Supplementary Fig. 6d). Western blot analysis detected accumulation of Fam3a-Myc-Flag within C2C12 myoblasts upon monensin treatment (Fig. 5c, d). Finally, we observed the presence of Fam3a-Myc-Flag in the media of transfected cells (Fig. 5e). Together, these findings demonstrate that Fam3a is secreted by myogenic cells.

Fam3a is a ubiquitously expressed cytokine-like protein[50], suggesting that it could also be produced by other muscle-resident cell types known to promote MuSC myogenic lineage progression, such as fibroadipogenic progenitors (FAPs) or macrophages (MPs)[51–53]. To address this question, we performed

quantitative real-time PCR (qPCR) analysis in muscle-resident cell types isolated from two highly myogenic contexts: postnatal muscle growth and adult tissue regeneration (Supplementary Fig. 7). During postnatal growth, we isolated mononucleated cells (MuSCs, MPs, and FAPs) and myofibers from P15 pups and 3-month-old C57BL6/J mice (Fig. 5f and Supplementary Fig. 7). Our results show that myofibers contain much higher levels of Fam3a transcript than mononucleated cells (Fig. 5f), suggesting that myofibers are the main producers of Fam3a in the skeletal muscle during postnatal growth and also in the adult skeletal muscle.

To assess Fam3a expression during adult skeletal muscle regeneration, we isolated mononucleated cells from 3-month-old male mice at three different time points: uninjured, 3 dpi, and 7 dpi (Fig. 5g and Supplementary Fig. 7). Three days after injury, both MPs and FAPs are the highest Fam3a-expressing cells at the population level (Fig. 5g). Seven days after injury, FAPs are the main Fam3a-expressing cells in the tissue (Fig. 5g), suggesting that FAPs may be a relevant source of Fam3a during skeletal muscle regeneration. Overall, our data suggest that Fam3a is a secreted factor that may mediate the coordination of the different muscle-resident cell types during myogenesis.

**Fam3a rescues myogenic commitment of Stat3 KO MuSCs.** To evaluate whether the secretion of Fam3a is relevant for its role on MuSC function, we performed rescue studies by adding Fam3a recombinant protein into the culture media. First, we used shRNA-mediated Fam3a loss of function (shFam3a). Addition of recombinant Fam3a increased the percentage of myogenin+ cells in shFam3a MuSCs cultured in both GM and DM (Fig. 6a and Supplementary Fig. 8a). Treatment did not affect the percentage of myogenin+ cells in control MuSCs (shCt) (Fig. 6a and Supplementary Fig. 8a), suggesting that control cells produce sufficient Fam3a levels to sustain their myogenic lineage progression and differentiation in culture. In a second set of experiments, we asked whether incubation with recombinant Fam3a was sufficient to rescue the defects in myogenic lineage progression and differentiation of Stat3 KO MuSCs in vitro. Indeed, addition of Fam3a protein into the media of Stat3 KO MuSCs rescued the percentage of myogenin+ cells up to the levels of control cells when cultured in GM (Fig. 6b), validating that it is a relevant Stat3 downstream effector. Treatment with recombinant Fam3a was not sufficient to rescue the deficit in myogenic differentiation of Stat3 KO MuSCs, suggesting that other downstream targets of Stat3 also play a relevant role in differentiation conditions (Supplementary Fig. 8b). Treatment did not affect the percentage of myogenin+ cells in control MuSCs (Fig. 6b and Supplementary Fig. 8b).

Our data demonstrate that both Stat3 and Fam3a promote mitochondrial function during MuSC activation. Thus we assessed whether treatment with Fam3a could also rescue the reduced mitochondrial respiration observed in Stat3 KO MuSCs. Incubation of Stat3 KO MuSCs with Fam3a recombinant protein for 72 h in growth conditions increased their basal and maximal mitochondrial respiration to levels similar to control cells (Fig. 6c, d) without affecting the ECAR (Fig. 6e). Consistent with our previous data, treatment with recombinant Fam3a did not affect mitochondrial respiration in control MuSCs (Supplementary Fig. 8c–e).

Finally, to demonstrate that Fam3a secretion also plays a relevant role during adult skeletal muscle regeneration in vivo, we performed rescue studies using Tmx-treated 4-month-old Pax7-CreER;Stat3f/f male mice and control littermates. Tibialis anterior muscles of these mice were injured with barium chloride, and 1 day after injury we delivered Fam3a recombinant protein (or

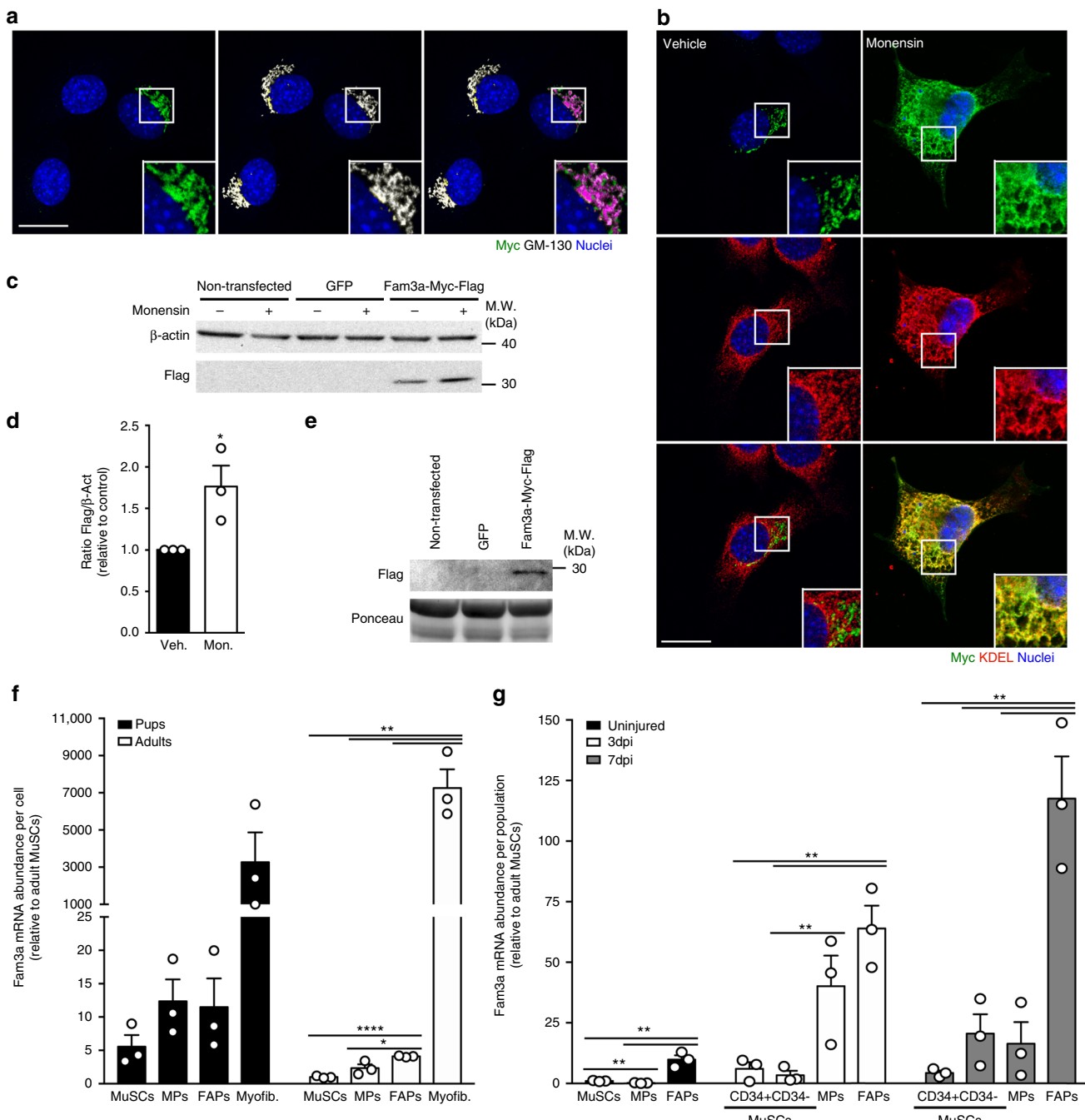

**Fig. 5** Fam3a is secreted by myogenic cells. **a**, **b** Immunofluorescence analysis of C2C12 myogenic cells transfected with a construct to overexpress Fam3a-Myc-Flag. GM-130 is a cis-Golgi marker. KDEL is an endoplasmic reticulum marker. Purple or yellow colors mean colocalization. When indicated, cells were treated with Monensin (200 μM) for 4 h. Scale bars 20 μm. **c**, **d** Western blot analysis of non-transfected C2C12 myoblasts and C2C12 cells transfected with constructs to overexpress green fluorescent protein (GFP) or Fam3a-Myc-Flag. When indicated, cells were treated with Monensin (200 μM) for 4 h ($n = 3$ independent experiments). **e** Western blot analysis of the media from C2C12 cells transfected with a construct to overexpress Fam3a-Myc-Flag. **f** Fam3a mRNA abundance per cell in freshly isolated muscle stem cells (MuSCs), macrophages (MPs), fibroadipogenic progenitors (FAPs), and myofibers from C57BL6/J P15 pups ($n = 3$ biologically independent samples; 3 pups were pulled per sample) and 3-month-old adult male mice ($n = 3$ animals). **g** Fam3a mRNA abundance per population in freshly isolated MuSCs, MPs, and FAPs from uninjured, 3 dpi, and 7 dpi 3-month-old male mice ($n = 3$ animals). Data are represented as mean ± SEM (Student's $t$ test; *$p < 0.05$, **$p < 0.01$, ****$p < 0.0001$)

vehicle as control) by intramuscular injection (Fig. 6f). Vehicle-treated muscles from Pax7-CreER;Stat3[f/f] mice displayed a higher number of Pax7[+] cells 6 days after the initial injury compared to vehicle-treated muscles from control animals, as previously shown (Fig. 6g)[20]. Strikingly, treatment with recombinant Fam3a

of Pax7-CreER;Stat3[f/f] injured muscles normalized the number of Pax7[+] cells to the levels of control mice (Fig. 6g). Moreover, treatment with recombinant Fam3a significantly increased the number of myogenin[+] cells in injured Pax7-CreER;Stat3[f/f] muscles (Fig. 6h). Overall, this data indicates that Fama3a is a

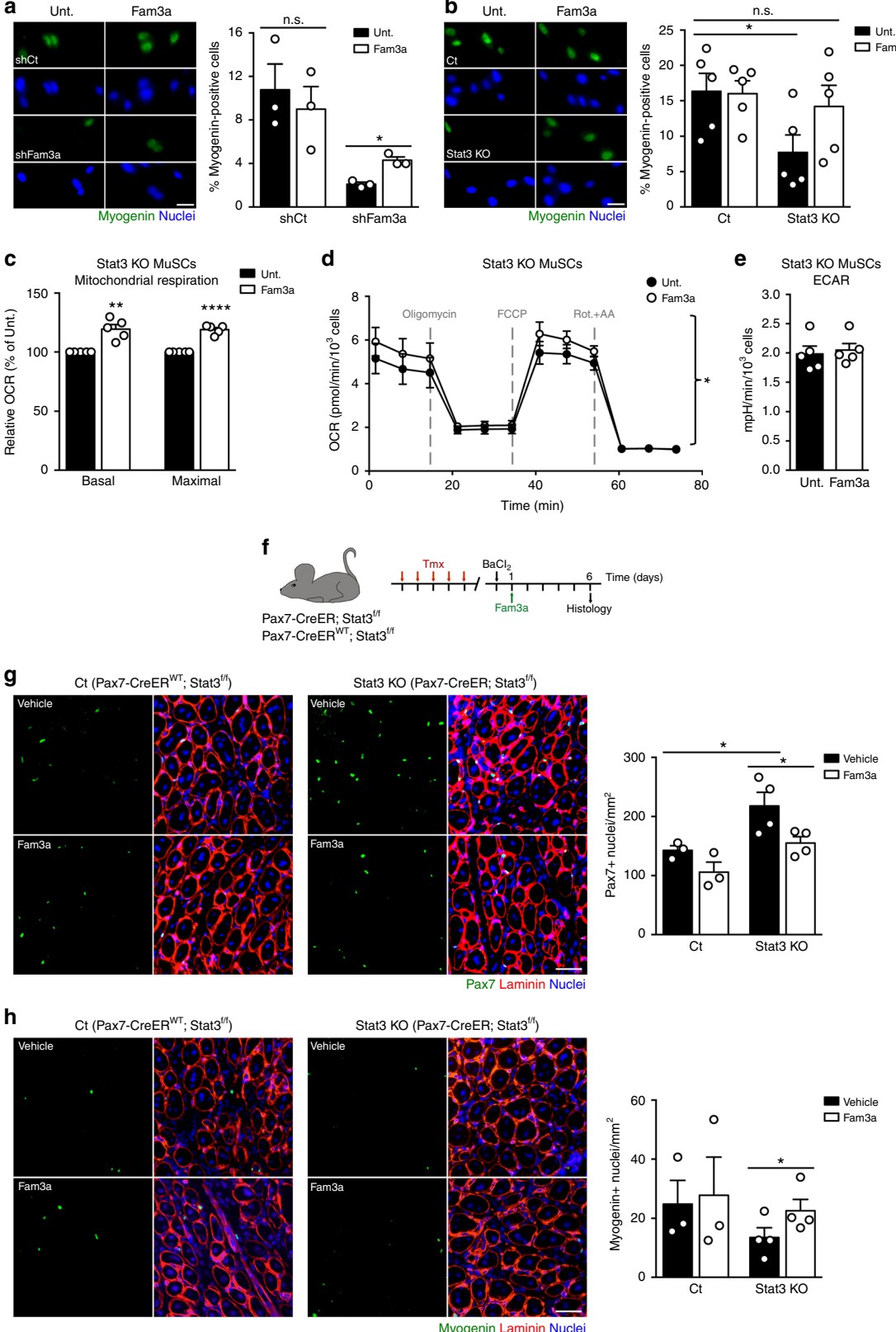

major downstream effector of Stat3 in MuSCs that promotes myogenic commitment in vivo.

## Discussion

Understanding the mechanisms that regulate stem cell fate choices is one of the goals of regenerative medicine in order to develop efficient therapeutic approaches, and stem cell metabolic reprogramming is emerging as a strategy with the potential to improve tissue repair[1,2]. Common to several stem cell compartments, MuSC commitment and differentiation requires the induction of oxidative metabolism[2,3,9–12]. For instance, the induction of mitochondrial function is required for neuronal

**Fig. 6** Fam3a rescues impaired commitment of Stat3 knockout (KO) muscle stem cells (MuSCs) in vitro and in vivo. **a** Immunofluorescence analysis and quantification of myogenin expression in MuSCs infected with shCt or shFam3a coding lentiviruses and cultured in growth conditions for 72 h in the presence or absence of recombinant Fam3a (1000 ng/ml) (n = 3 independent experiments). Scale bar 25 μm. **b** Immunofluorescence analysis and quantification of myogenin expression in control and Stat3 KO MuSCs cultured in growth conditions for 72 h in the presence or absence of recombinant Fam3a (1000 ng/ml) (n = 5 independent experiments). Scale bar 25 μm. **c–e** Measurement of the oxygen consumption rate (OCR) and the extracellular acidification rate (ECAR) of Stat3 KO MuSCs cultured in growth conditions for 3 days in the presence or absence of recombinant Fam3a (1000 ng/ml) (n = 5 independent experiments). The Cell Mito Stress Test was performed using the Seahorse XFp Analyzer. **f** Scheme of the experimental design for the intramuscular Fam3a treatment in vivo. **g, h** Immunofluorescence analysis and quantification of Pax7- and myogenin-positive cells in transversal sections of the tibialis anterior muscles of Pax7-CreER[WT];Stat3[f/f] and Pax7-CreER;Stat3[f/f] male mice 6 days after barium chloride injury. Vehicle or 1 μg of Fam3a recombinant protein was injected intramuscularly 1 day after inducing injury (n = 3–4 animals). Scale bar 25 μm. Data are represented as mean ± SEM (Student's *t* test or two-way analysis of variance; *p < 0.05, **p < 0.01, ****p < 0.0001)

differentiation[6,7], the osteogenic and adipogenic differentiation of mesenchymal stem cells[1,8], myeloid differentiation of hematopoietic stem cells[4,5], and cardiac differentiation[54].

Here we used MuSCs as a model to study the factors that promote mitochondrial activity during stem cell differentiation. Previous studies performed in cultured myoblasts proved that mitochondrial respiration promotes myogenic differentiation[32–36]. Consistent with this data, we have shown that inhibition of oxidative phosphorylation impairs MuSC myogenic commitment, and we have characterized the Stat3–Fam3a axis as an inducer of mitochondrial respiration that promotes myogenic differentiation. Indeed, transcriptomic analysis of early activated Stat3-ablated MuSCs showed reduced mitochondrial respiration before any alteration in the expression of the myogenic markers occur, suggesting that the defects in mitochondrial function precedes the impairment in myogenic commitment of Stat3 KO MuSCs. These findings further demonstrate that Stat3 translates microenvironmental cues to changes in stem cell metabolism.

Stat3 has been previously shown to regulate cell metabolism in a complex manner depending on posttranslational modifications, cellular localization, and cell type, with the current view that nuclear Stat3 promotes glycolysis, while mitochondrial Stat3 promotes oxidative phosphorylation[22–24]. Our data show that nuclear Stat3 can also promote mitochondrial function by regulating the expression of nuclear-encoded genes in MuSCs. Indeed, here we provide evidence that Stat3 directly promotes the expression of the secreted protein Fam3a as one downstream target that increases mitochondrial respiration in MuSCs. We and others previously showed that Stat3 induces MuSC commitment by regulating MyoD expression[19,20], and a recent report indicated that MyoD is required to maintain the respiratory capacity of the mitochondria in C2C12 myotubes and adult skeletal muscles by regulating the expression of nuclear-encoded genes[10]. Here we demonstrate that both Stat3 and MyoD are able to induce *Fam3a* expression in myogenic cells by directly binding to its promoter. Additionally, ChIP-seq data showed that MyoD binds to the *Fam3a* promoter in both mouse and human models of myogenic differentiation, suggesting that the regulation of *Fam3a* expression is conserved between species. Overall, this data points to the existence of a regulatory network, which includes Stat3 and MyoD, that aims at progressively increasing *Fam3a* expression and oxidative metabolism in myogenic progenitors to ensure efficient differentiation.

Fam3a biological function is still poorly understood and a potential role of Fam3a in the skeletal muscle has not been previously reported. Here we show that Fam3a is required for proper MuSC myogenic lineage progression and skeletal muscle development in vivo. Indeed, our data suggest that Fam3a genetic ablation affects the dynamics of the myogenic progenitors during development. Specifically, Fam3a absence results in reduced number of myogenin[+] cells and increased number of uncommitted MuSCs (Pax7[+]), suggesting that there is a delay in the myogenic lineage progression of MuSCs during the skeletal muscle development of Fam3a-null mice. Interestingly, our studies showed that partial Fam3a loss of function is sufficient to impair MuSC myogenic commitment, suggesting that there is a requirement of a minimum level of Fam3a to ensure proper myogenic lineage progression.

We further demonstrate that Fam3a is a cytokine-like protein secreted by myogenic cells. Our data show that myofibers are the major expressers of Fam3a in the skeletal muscle. This is consistent with the progressive increase in Fam3a expression during myogenic differentiation and suggests that the Fam3a secreted by myogenic cells plays a major role during skeletal muscle growth and regeneration by acting in a paracrine and/or autocrine manner. We observed that Fam3a is also expressed by other muscle-resident cells such as FAPs and MPs. FAPs and MPs are known to support myogenic lineage progression[51–53], which suggests that Fam3a could also be produced by these other cell types and contribute to the differentiation of myogenic progenitors. We believe that this could be especially relevant to promote tissue repair in the context of adult skeletal muscle regeneration, where there is a transient increase in the abundance of MPs and FAPs in the skeletal muscle. However, during muscle tissue growth and homeostasis, myogenic cells are the major producers of Fam3a in the tissue microenvironment.

Consistent with previous reports showing that Fam3a stimulates mitochondrial respiration in hepatocytes, neuronal cells, and vascular smooth muscle cells[25–27], we demonstrate that Fam3a plays a similar role in MuSCs. Genetic ablation of Fam3a reduces MuSC mitochondrial respiration, while treatment with recombinant Fam3a rescues the defects in mitochondrial respiration in Stat3-ablated MuSCs. Previous studies showed that inhibition of mitochondrial respiration in myogenic cells specifically compromises the induction of myogenin expression during myogenic differentiation and does not affect the expression of other myogenic markers such as MyoD[34–36], which is equivalent to the phenotype we observe in our Fam3a loss-of-function studies. We provided evidence that Fam3a does not affect mitochondrial biogenesis, content, or the expression levels of different subunits of the electron transport chain complexes. However, the molecular mechanism by which Fam3a stimulates mitochondrial respiration in MuSCs is still unknown. Fam3a overexpression has been reported to induce Akt and AMPK activation in different cell types[25–27], and as both pathways have been shown to regulate MuSC metabolism and to promote myogenic commitment and differentiation[9,11,55], future studies should characterize the relationship between Fam3a and these signaling networks in MuSCs.

Overall, our findings indicate that Fam3a is a Stat3 and MyoD downstream target that promotes mitochondrial respiration and differentiation of MuSCs. Identification and characterization of regulators of MuSC metabolism is relevant for therapeutic

purposes as they can be utilized to modulate the balance between MuSC self-renewal and differentiation, improve tissue repair in diseased conditions with deficient skeletal muscle regenerative capacity, and ameliorate muscle wasting.

## Methods

**Animal procedures.** All protocols were approved by the Sanford Burnham Prebys Medical Discovery Institute Animal Care and Use Committee and by the Italian Ministry of Health, the National Institute of Health (IIS), the Santa Lucia Foundation (Rome) Animal Care and Use Committee. The study is compliant with all relevant ethical regulations regarding animal research. Mice were housed according to institutional guidelines, in a controlled environment at a temperature of 22 °C ± 1 °C, under a 12-h dark–light period and provided with standard chow diet and water ad libitum. Male Pax7-CreER;Stat3$^{f/f}$ mice and control Pax7-CreER$^{WT}$;Stat3$^{f/f}$ littermates (between 3- and 7-month-old) were used. Pax7-CreER;Stat3$^{f/f}$ mice were maintained in C57BL/6J background[20]. Fam3a KO mice were generated in a C57BL/6J background, and F0 generation E15 male embryos and P0 male pups were analyzed. Adult 3-month-old Fam3a KO mice were also used in this study. MyoD−/− mice were maintained in a B6;129 genetic background[41,42], and 3-month-old male mice were used. Young (3–4-month-old) male C57BL/6J mice were purchased from Jackson Laboratories. Day 15 C57BL6/J pups were obtained from the C57BL/6J colony maintained by the SBP Animal Facility. Mice within the same genotype were randomized in all the studies.

Three-month-old Pax7-CreER;Stat3$^{f/f}$ mice and control Pax7-CreER$^{WT}$;Stat3$^{f/f}$ littermates were treated with Tmx (i.p., 0.1 mg/g body weight, cat.#T5648, Sigma) for 5 consecutive days. Tmx was resuspended in corn oil. Experiments were performed after at least 2 weeks from the last Tmx injection (mice were between 4- and 7-month-old).

To induce skeletal muscle injury, mice were anesthetized by 1–4% L/min O$_2$ isoflurane (cat.#502017, MWI Vet Supply) inhalation, and tibialis anterior muscles were injured by intramuscular injection of 50 μl of barium chloride (1.2% w/v, cat.#202738, Sigma). Tissues were harvested at the indicated time points for FACS or histological analysis.

For the Fam3a in vivo treatment, 1 μg of Fam3a recombinant protein (or vehicle as control) was injected intramuscularly into the tibialis anterior muscles of the mice 1 day after inducing barium chloride injury. Histological analysis was performed 6 days after the initial injury.

**Generation of Fam3a KO mice.** CRISPR/Cas9 gene editing to generate Fam3a mutant embryos was performed. A single guide RNA (sgRNA) to target inside exon 2 was designed using Guidescan.com to ensure maximum specificity and cutting efficiency (sgRNA sequence: CTAGTCACCATCCTCCTAGG). DNA template for the sgRNA was generated by PCR amplification (Q5 DNA Polymerase; cat. #M0491S, New England BioLabs) of ssDNA ultramer oligonucleotide (Integrated DNA Technologies); sgRNA was transcribed from this template using the HiScribe T7 High Yield RNA Synthesis Kit (cat.#E2040S, New England Biolabs), treated with DNAse I (cat.#M0303S, New England Biolabs), and purified using the Megaclear Kit (cat.#AM1908, Invitrogen). For mouse zygote injections, 50 ng/μl Cas9 mRNA (Life Technologies) and 20 ng/μl sgRNA were combined in IDTE buffer (IDT). Fertilized oocytes were collected from 3- to 4-week-old superovulated C57BL/6J females (prepared by injecting 5 IU each of pregnant mare serum gonadotropin and human chorionic gonadotropin), then transferred into M2 medium (Millipore), and injected with the Cas9 mRNA/sgRNA solution into the cytoplasm. Injected embryos were then re-implanted into recipient pseudo-pregnant ICR female mice. Some of the implanted females were sacrificed 15 days after re-implantation for E15 embryos analysis. Tail DNA was collected for genotyping by PCR (cat.#Bio-21126, Bioline MyTaq Extract Kit) (see Supplementary Table 1 for primer sequences) followed by Sanger sequencing to assess for mutations. Fam3a KO mice that displayed in-frame indels were excluded from the study. Genotyping to identify the gender of the embryos and P0 pups was performed by amplifying fragments with different lengths from the X and Y chromosomes[56].

**Mononucleated cell isolation from adult skeletal muscle.** Mononucleated cells were isolated from tibialis anterior, gastrocnemius, and quadriceps muscles by FACS[57]. Tissues were minced and subjected to an initial enzymatic digestion (700 units/ml collagenase, cat.#17101015, Life Technologies) for 90 min. After a second enzymatic digestion (100 units/ml collagenase and 2 units/ml dispase (cat. #17105041, Life Technologies)) for 30 min, the cell suspension was filtered through a 70-μm nylon filter.

For the experiments where only MuSCs were isolated, samples were then incubated with the following biotinylated rat antibodies: CD45 (clone 30-F11, cat. #553078, BD Biosciences, 1/400), CD11b (cat.#553309, BD Biosciences, 1/150), CD31 (cat.#5011513, eBioscience, 1/150), and Sca1 (clone E13-161.7, cat.# 553334, BD Biosciences, 1/150). Streptavidin microbeads (cat.#130-048-101, Miltenyi) were then added to the cells together with the α7-integrin–phycoerythrin antibody (clone R2F2, cat.#67-0010-05, Ablab, 1/150), CD34–Alexa647 antibody (clone RAM34, cat.#560230, BD Biosciences, 1/50), and Streptavidin APC-Cy7 (cat.

#554063, BD Biosciences, 1/150). Magnetic depletion of biotin-positive cells was performed before fractionating the (CD45$^−$CD11b$^−$CD31$^−$Sca1$^−$) α7-integrin $^+$CD34$^+$ population by flow cytometry (BD FACS Aria) and purity check.

For the experiments isolating the three populations of mononucleated cells (MuSCs, MPs, FAPs), samples were incubated with the following antibodies: CD45 (clone 30-F11, cat.#553078, BD Biosciences, 1/400), CD11b (cat.#553309, BD Biosciences, 1/150), CD31 (cat.#5011513, eBioscience, 1/150), F4/80-FITC (cat. #114801-82, eBioscience, 1/150), α7-integrin–phycoerythrin antibody (clone R2F2, cat.#67-0010-05, Ablab, 1/150), CD34–Alexa647 antibody (clone RAM34, cat. #560230, BD Biosciences, 1/50), Sca1-Pacific Blue (clone E13-161.7, cat.# 122520, Biolegend, 1/150), and Streptavidin APC-Cy7 (cat.#554063, BD Biosciences, 1/ 150). The following populations were isolated by flow cytometry (BD FACS Aria): MuSCs (CD45$^−$CD11b$^−$CD31$^−$Sca1$^−$α7-integrin$^+$CD34$^+$) MPs (CD45$^+$CD11b $^+$CD31$^+$F4/80$^+$) and FAPs (CD45$^−$CD11b$^−$CD31$^−$Sca1$^+$α7-integrin$^−$CD34$^+$). Furthermore, for the two time points after injury, we isolated two different populations of MuSCs: CD45$^−$CD11b$^−$CD31$^−$Sca1$^−$α7-integrin$^+$CD34$^+$ cells and CD45$^−$CD11b$^−$CD31$^−$Sca1$^−$α7-integrin$^+$CD34$^−$ progenitors.

**Mononucleated cell isolation from P15 pup skeletal muscle.** MuSCs, MPs, and FAPs were isolated from the hind limb skeletal muscles of 15 days old (P15) C57BL/6J pups by FACS[58]. Briefly, hind limb muscles were harvested, minced, and enzymatically digested for 45–50 min (45 units/ml Collagenase A, Roche; 2.4 units/ ml Dispase I, Roche) for 45–50 min at 37 °C in a rotating water bath. Next, cell suspension was filtered through 70-μm nylon filter and incubated with the primary antibodies listed in the previous section. After incubation, cells were washed, resuspended in FACS Buffer, and filtered through 40-μm strain filter. Flow cytometry (BD FACS Aria) was performed to isolate MuSCs (CD45$^−$CD11b$^−$CD31$^−$ Sca1$^−$α7-integrin$^+$CD34$^+$), MPs (CD45$^+$CD11b$^+$CD31$^+$F4/80$^+$), and FAPs (CD45$^−$CD11b$^−$CD31$^−$Sca1$^+$α7-integrin$^−$CD34$^+$). FACS data were analyzed in FlowJo (version 10.0.4; FlowJo LLC).

**Single myofiber isolation.** Single myofibers were isolated from the gastrocnemius/ soleus muscles from adult and P15 mice[59]. In brief, muscles were harvested and subjected to enzymatic dissociation (700 U/ml collagenase type II, Gibco cat. #17101-015) for 60 min at 37 °C in a shaking water bath. Dissociated single myofibers were manually collected under a dissection microscope and gently washed three times with Dulbecco's modified Eagle's medium (DMEM) containing 10% horse serum to remove debris and interstitial cells. Single myofibers were then pooled and lysed with QIAzol lysis reagent (Qiagen cat.#79306) for subsequent RNA extraction and qPCR analysis.

**RNA sequencing.** MuSCs were isolated from Pax7-CreER;Stat3$^{f/f}$ and Pax7-CreER$^{WT}$;Stat3$^{f/f}$ mice. Total RNA was isolated according to the manufacturer's instructions with DNAse digestion (cat.#217084, Qiagen). RNA quality was verified by the Agilent 2100 Bioanalyzer. RNA-seq was performed using an Illumina HiSeq 2500 instrument by the Analytical Genomics and Bioinformatics Core Facility (SBP, Lake Nona). Up to 1 μg of total RNA was used to prepare standard Illumina single read True-seq libraries and library quality was controlled by using Agilent Bioanalyzer, Qubit, and KAPA qPCR. The KAPA primers target Illumina P5 and P7 flow-cell oligo sequences provide an accurate indication of the quantity of a given library. Spiking control PhiX DNA was applied to monitor the run metrix. The Illumina machine built-in software provided read quality scores ($Q > 30$). The raw reads (FastQ files) from Illumina HiSeq2500 were aligned to the mouse genome (mm10 assembly) using TopHat2 splice-aware aligner with RefSeq annotations and the "—no-novel- juncs" option[60]. Ambiguous reads that mapped to >1 region in the genome and reads with MAPQ score <10 were removed. Expectation–Maximization approach was used to estimate transcript abundance[61], and the raw read counts and normalized read counts (RPKM: reads per kilobase per million mapped reads) were obtained[62]. Normalization between samples was also performed[63]. Gene differential expression analysis was done using the generalized linear model likelihood ratio test implemented in the edgeR software[64]. Transcripts detected in at least one sample (RPKM > 1), fold change >2, and $p$ value < 0.05 were considered as significant differential expression. GSEA with pre-ranked method was used to perform pathway analysis[29]. Normalized gene RPKM values were averaged within groups for heat map generation.

**Cell culture procedures.** All cells were cultured in incubators at 37 °C and 5% CO$_2$. Freshly isolated MuSCs were plated on tissue culture plates coated with laminin (cat.#11243217001, Roche) and maintained in GM (45% DMEM, 40% F10, 15% fetal bovine serum (FBS)). Myogenic differentiation was induced with DMEM and 2% horse serum for 2 or 3 days. MuSC proliferation was measured by 5-ethynyl-2′-deoxyuridine (EdU) incorporation (5 ng/μl, cat.#A10044, Life Technologies) for 4 h at the end of the culture. MuSCs were treated with CCCP (12.5 μM when cultured in GM and 5 μM when cultured in DM) (cat.#C2759, Sigma) for 24 h when indicated. MuSCs were treated with recombinant FAM3A (1000 ng/ml, cat.#TP303495, OriGene) for 72 h when indicated.

C2C12 myogenic cells were obtained from ATCC (crl-1772) and we did not perform further authentication. C2C12 cells were grown in High Glucose DMEM supplemented with FBS (10%). Myogenic differentiation was induced with High Glucose DMEM and 2% horse serum for 5 days. C2C12 myogenic cells were treated with recombinant IL-6 (100 ng/ml, cat.# 216-16, Peprotech) for 24 h when indicated. C2C12 cells were treated with 200 μM Monensin (cat.#M5273, Sigma) for 4 h when indicated.

HEK-293 cells were obtained from ATCC (crl-1573) and grown in High Glucose DMEM supplemented with 10% FBS. We did not perform further authentication.

IMR90 cells were obtained from Coriell (I90-10) and grown in Eagle's minimal essential medium (EMEM) (ATCC) supplemented with 10% FBS. We did not perform further authentication.

**Immunofluorescence in muscle sections.** Muscle tissues were dissected and embedded in optimal cutting temperature compound (cat.#4583; Tissue-Tek; Sakura Finetek), snap-frozen in isobutane and liquid nitrogen, and sectioned at 10-μm thickness. Muscle sections were fixed with 4% paraformaldehyde (PFA) for 15 min. For eMyHC staining, sections were fixed with acetone for 10 min at −20 °C. After three washes with phosphate-buffered saline (PBS) for 10 min, sections were permeabilized and blocked in 20% Goat Serum (cat.#16210-072, Life technologies) and 0.3% Triton (cat.#H5142, Promega) in PBS for 1 h. For Pax7 staining, antigen retrieval was performed and sections were post fixed with 4% PFA. Incubation with the primary antibodies was performed overnight (O/N) at room temperature (RT). The following primary antibodies were used: mouse anti-Pax7 (cat.#Pax7-c, Developmental Studies Hybridoma Bank (DSHB), 1/100 dilution), rabbit anti-Laminin A1 (cat.#L9393, Sigma, 1/100 dilution), rat anti-laminin B2 (cat.#05-206, Millipore, 1/50 dilution), mouse anti-myogenin (cat.#556358, BD Biosciences, 1/100), and mouse anti-eMyHC (cat.#F1.652, DSHB, 1/100). Alexa-conjugated secondary antibodies (Invitrogen, 1/250 dilution) were used. Nuclei were counterstained with Hoechst 33342 (cat.#H3570, Invitrogen). Images of muscle transverse sections were acquired using: the Inverted IX81 Olympus Compound Fluorescence Microscope with ×10 or ×20 magnification lenses, Color/monochrome cooled CCD camera - Spot RT3, and MetaMorph 7.11 Software (UIC, Molecular Devices) or the Leica TCS SP8 confocal microscope with the ×20 magnification lens and LAS X software. Images were composed and edited using Photoshop CS6 (Adobe).

**Immunofluorescence in cultured cells.** Cells were fixed with 1.5% PFA for 15 min and washed three times with PBS for 10 min. For Stat3 and S727 pStat3 stainings, cells were fixed with 4% PFA for 15 min and with cold methanol (−20 °C) for 10 min. Next, cells were permeabilized and blocked in 20% Goat Serum (cat.#16210-072, Life Technologies) and 0.3% Triton (cat.#H5142, Promega) in PBS for 1 h. Incubation with the primary antibodies was performed O/N at RT. The following primary antibodies were used: mouse anti-Pax7 (cat.#Pax7-c, Developmental Studies Hybridoma Bank (DSHB), 1/100 dilution), rabbit anti-Myod1 (cat.#sc760, Santa Cruz, 1/30 dilution), mouse anti-myogenin (cat.#556358, BD Biosciences, 1/100 dilution), mouse anti-Myosin Heavy Chain (cat.#Mf20-c, DSHB,1/50 dilution), rabbit anti-Ki67 (cat.#ab15580, Abcam, 1/100 dilution), rabbit anti-Myc tag (cat. #ab9106, Abcam, 1/100 dilution), rat anti-KDEL (cat.#ab50601, Abcam, 1/150 dilution), rabbit anti-Tomm20 (cat.# NBP1-81556, Novus Biologicals, 1/100 dilution), mouse anti-Flag (cat.#F3165, Sigma, 1/500 dilution), mouse anti-GM130 (cat.#610823, BD Biosciences, 1/150 dilution), mouse anti-Stat3 (cat.#9139, Cell Signaling, 1/100 dilution), and rabbit anti-pStat3 (S727) (cat.#9134, Cell Signaling, 1/50 dilution). Alexa-conjugated secondary antibodies (Invitrogen, 1/250 dilution) were used. Nuclei were counterstained with Hoechst 33342 (cat.#H3570, Invitrogen). EdU staining was performed following the manufacturer's instructions (cat. #C10340, Life Technologies). Images of cultured cells were acquired using the Inverted IX81 Olympus Compound Fluorescence Microscope with ×10 or ×20 magnification lenses, Color/monochrome cooled CCD camera - Spot RT3, and MetaMorph 7.11 Software (UIC, Molecular Devices). For the subcellular localization studies, the Leica TCS SP8 confocal microscope with the ×20 and ×40 magnification lens and LAS X software was used. For colocalization studies, Pearson's correlation coefficient was calculated using Coloc2 (with Costes threshold correction) in Fiji[65]. Corrected Total Cell Fluorescence for Tomm20 was calculated using Fiji[65]. Images were composed and edited using Photoshop CS6 (Adobe).

**Lentiviral infection.** Lentiviruses were purchased from the MISSION shRNA library (Sigma) as part of an agreement with the Sanford Consortium of Regenerative Medicine. Tissue culture plates were coated with laminin (20 μg/ml) (cat. #11243217001, Roche) for 45 min at 37 °C, followed by RetroNectin (15 μg/ml) (cat.#T100A, Takara) for 2 h at RT. Plated MuSCs were infected with lentiviruses coding for an shFam3a or non-mammalian control (shCt) in GM supplemented with 8 ng/ml polybrene (cat.#TR-1003-6, Millipore) O/N at 37 °C and 5% CO2.

**Transient transfection.** C2C12 cells were transfected with a Fam3a overexpression plasmid (cat.#MR202771, OriGene) and/or a green fluorescent protein overexpression plasmid (cat.#13031, Addgene) using Lipofectamine™ 3000 (cat.

#L3000001, Invitrogen) and following the manufacturer's instructions. Cells were transfected with 1 μg DNA.

**IMR90 myogenic conversion.** IMR90 cells were electroporated using the Neon Transfection System (Invitrogen, MPK5000, MPK10025) with helper plasmid and epB-Puro-TT containing or not murine Myod1 cDNA. Cells were then selected with 2 μg/ml of puromycin dihydrochloride (MP Bio). When cells were 60% confluent, Myod1 was induced with 200 ng/ml doxycycline (Sigma) in GM for 24 h and cells were collected for the GM point. When cells were 95–100% confluent, Myod1 was induced with 200 ng/ml doxycycline (Sigma) in GM for 24 h and then cells were differentiated in EMEM supplemented with 2% horse serum (Gibco), 1% ITS (Sigma), and 200 ng/ml doxycycline for 3 days for the DM time point. Media with doxycycline was refreshed every 2 days.

**Luciferase reporter assay.** The Fam3a regulatory region (−3000 to +100 bp) was cloned into a pGL3 Luciferase plasmid (Fam3a-Luc). The following vectors were used: Fam3a-Luc, pRL Renilla Luciferase Control Reporter Vector (Promega), pcDNA3-Stat3[66], and pcDNA3-myc-MyoD[41]. HEK293 cells were transfected with jetPRIME (cat.#114-07, Polyplus transfection) following the manufacturer's instructions. Luciferase and Renilla activities were measured 48 h after transfection using the Dual Luciferase Reporter Assay System (cat.#E1910 Promega) following the manufacturer's instructions.

**Gene expression analysis.** RNA was extracted from cells using the Qiagen miR-Neasy Micro Kit (cat.#217084, Qiagen) following the manufacturer's protocol. Total RNA was quantified with a Qubit™ 3 Fluorometer (cat.#Q33216, Thermo Scientific). First-strand cDNA was synthesized from total RNA using the SuperScript™ VILO™ cDNA Synthesis Kit (cat.#11754050, Invitrogen) following the manufacturer's instructions. cDNA was used in qPCR using the LightCycler® 96 Instrument. qPCR reactions consisted of Power SYBR™ Green PCR Master Mix (cat.# 4367659, Life Technologies), 250 nM forward and reverse primers, and 0.5 ng of cDNA. All measurements were normalized to Rplp0 expression using the $2^{-\delta\Delta CT}$ method. The sequences of the primers used are in Supplementary Table 1.

**Western blotting.** Total protein extracts for western blot analyses were obtained by homogenizing C2C12 cells in RIPA buffer (50 mM Tris-HCl pH = 7.4, 150 mM NaCl, 1 mM EDTA, 1.0% NP-40, 0.5% sodium deoxycholate, 0.1% sodium dodecyl sulfate (SDS)) supplemented with protease inhibitors (cat.#11836153001, Sigma) and phosphatase inhibitors (cat.#4906837001, Sigma) cocktails. Cell debris was removed by centrifugation at $700 \times g$ for 10 min and 4 °C and collecting the supernatant. Protein concentration was determined using the Pierce™ BCA Protein Assay Kit (cat.#23225, Thermo Scientific). Total protein extracts (20 μg) were resolved in NuPAGE™ 4–12% Bis-Tris Protein Gels (cat.#NPO335BOX, Invitrogen) by electrophoresis and then transferred to polyvinylidene difluoride membranes (cat.#1620177, BioRad). After the transference, membranes were blocked with 1× PBS and 0.1% Tween-20 (PBST) with 5% w/v non-fat milk at RT for 1 h, followed by an O/N incubation with the diluted antibodies in blocking buffer at 4 °C with gentle shaking. After washing with PBST, membranes were incubated at RT for 1 h with a goat polyclonal anti-mouse IgG secondary antibody conjugated to horseradish peroxidase (cat.#1705047, BioRad). Finally, proteins were detected by enhanced chemiluminescence (Pierce™ ECL Western Blotting Substrate, cat. #32106, Thermo Scientific) followed by exposure to film. The following primary antibodies were used: mouse anti-beta actin (cat.#ab20272, Abcam, 1/1000), mouse anti-Flag (cat.#F3165, Sigma, 1/1000), and total OXPHOS Rodent WB antibody cocktail (cat.#ab110413, Abcam, 1/1000).

For cell culture media western blot, C2C12 cells were cultured in media without serum for 30 h. After incubation, the media was centrifuged at $3200 \times g$ for 10 min and 4 °C and the supernatant was collected. Then the supernatant was placed in the Amicon® Ultra-2 centrifugal filter units (cat.#Z740164, Sigma) and concentrated following the manufacturer's instructions. After concentrating the media, samples were resolved as described above.

Unprocessed images for western blots included in this manuscript are available in Supplementary Fig. 9.

**C2C12 myotubes' ChIP-seq analysis.** ChIP-seq data for MyoD binding in C2C12-derived myotubes was retrieved from SRA with accession number SRX328691[40]. Reads were aligned to the UCSC mm10 build of the mouse genome using Bowtie (v 0.12.7)[67] with parameters -S -t -a -m 1 –best –strata –chunkmbs 200 and filtered with BEDtools (v2.26.0)[68] according to the original publication. Alignments were converted to bigwig format with RSeQC (v2.6.3)[69] and visualized in IGV (v2.3.98)[70].

**Chromatin immunoprecipitation.** C2C12 myoblasts were crosslinked with 1% formaldehyde in TBS for 15 min at RT. Formaldehyde was quenched by 0.125 mM Glycine for 5 min at RT. Crosslinked cells were quickly washed with cold TBS, and collected in cold TBS containing 1 mM phenylmethanesulfonylfluoride (PMSF; cat. #93482, Sigma), protease inhibitors (cat.#11836153001, Sigma), and phosphatase

inhibitors (cat.#4906837001, Sigma). After centrifugation at $400 \times g$ for 8 min and 4 °C, cell pellets were resuspended in ChIP lysis buffer containing 50 mM Tris-HCl pH = 8.0, 150 mM NaCl, 5 mM EDTA, 0.5% SDS, 0.5% NP-40, 1 mM PMSF and protease and phosphatase inhibitors. Chromatin was sheared to an average DNA fragment length of 500 bp using Misonix3000 sonicator. Cell lysates were diluted five times with the ChIP lysis buffer lacking SDS to a final concentration of 0.1% SDS. Samples were centrifuged and the DNA concentration of soluble chromatin was determined with a Qubit$^{TM}$ 3 Fluorometer (cat.#Q33216, Thermo Scientific). Thirty μg of DNA was used for immunoprecipitation with 5 μg of anti-Stat3 antibody (cat.#sc-482, Santa Cruz), anti-H3K27Ac antibody (cat.#39133, Active Motif), and an unspecific normal rabbit IgG as a control (cat.#sc-2027, Santa Cruz). After O/N incubation of the chromatin with the antibodies at 4 °C, the immuno-complexes were captured with 50 μl of Protein A magnetic beads (cat.#10001D, Invitrogen) (beads were pre-blocked with 5% IgG-free bovine serum albumin (cat. #001-000-161, Jackson ImmunoResearch Laboratories)) for further 4 h at 4 °C. Protein A-bound immunocomplexes were washed four times with buffer containing 50 mM Tris-HCl pH = 8.0, 150 mM NaCl, 5 mM EDTA, 0.1% SDS, 1% NP-40, 0.5% Sodium Deoxycholate, 1 mM PMSF, and protease and phosphatase inhibitors cocktails, followed by one wash with a buffer containing 100 mM Tris-HCl pH = 8.0, 250 mM LiCl, 5 mM EDTA, 1% NP-40, 1% Sodium Deoxycholate, and protease and phosphatase inhibitor cocktails. After one final wash with TE buffer, immunocomplexes were eluted from the beads and crosslinking was reversed by incubation for 5 h at 65 °C (800 rpm shaking) with TE buffer containing 1% SDS. After 0.2 mg/ml Proteinase K (cat.#P8107S, New England Bio-Labs) treatment of samples, DNA from immunoprecipitated samples as well as DNA from 10% input was purified by phenol and chloroform extraction and using the QIAquick PCR Purification Kit (cat.#28106, Qiagen) following the manu-facturer's instructions. In all, 1/30 of the purified DNA was analyzed by qPCR using the Power SYBR™ Green PCR Master Mix (cat.#4367659, Life Technologies). Primers used are listed in Supplementary Table 1. The ChIP signal was evaluated by calculating the amount of immunoprecipitated DNA relative to the input DNA (percentage of input) and relative to control conditions.

**IMR90 MyoD ChIP-seq.** Cells were fixed in 1% formaldehyde in PBS for 15 min at RT. Formaldehyde was then quenched with 125 mM Glycine for 5 min at RT. Cells were washed in PBS and harvested in PBS supplemented with 1 mM PMSF and protease inhibitors. Dry cell pellet was stored at −80 °C. Nuclei were then extracted and then lysed in lysis buffer containing 50 mM Tris-HCl, pH 8.0, 150 mM NaCl, 5 mM EDTA, pH 8.0, 0.5% SDS, 0.5% NP-40, 1 mM PMSF, and a protease inhibitor. Chromatin was sheared with sonicator (ColeParmer, Misonix 3000) to an average DNA fragment length of 200–500 bp. Chromatin was then diluted five times in lysis buffer without SDS. DNA amount was measured with the Qubit (Invitrogen Q32854). DNA was immunoprecipitated with 2 μg of rabbit anti-Myod1 (cat.#sc-760, Santa Cruz) O/N at 4 C. The immunocomplexes were captured with protein A magnetic beads (Life Technologies) for 3–4 h at 4 °C. After four washes with buffer containing 50 mM Tris-HCl, pH 8.0, 150 mM NaCl, 5 mM EDTA, pH 8.0, 0.1% SDS, 1% NP-40, and 0.5% sodium deoxycholate, one wash with a buffer containing 250 mM LiCl, 100 mM NaCl, 5 mM EDTA, pH 8.0, 1% NP-40, and 1% sodium deoxycholate, and two washes with TE buffer (10 mM Tris-HCl pH = 8, 1 mM EDTA), chromatin was then eluted and decrosslinked with 1% SDS in TE O/N at 65 °C 600 RPM rotation. Also, the input is decrosslinked with 1% SDS in TE O/N at 65 °C 600 RPM rotation. After 2-h digestion at 37 °C with 0.2 mg/ml Proteinase K, DNA was extracted with phenol/chloroform and ethanol precipitated O/N at −20 °C. Prior to sequencing, DNA was then suspended in mQ water. The DNA was then analyzed by qPCR calculating the amount of immunoprecipitated DNA relative to the input DNA (percentage of input).

For analysis, read quality was determined using FASTQC. Reads were mapped using bowtie2-2.0.5/bowtie2 to the female *Homo sapiens* hg19 genome with options: –very-sensitive-local. Over 85% of the reads successfully mapped. Duplicate reads were removed using samtools1.3. Peaks were called using macs2 2.1.1.20160309 with $q$ value < 0.01. Reads were extended based on the fragment size predicted with macs2.

**Seahorse Cell Mito Stress assay.** OCR and ECAR of cultured MuSCs were measured with the Seahorse XFp Analyzer following the manufacturer's instruc-tions. Briefly, freshly isolated MuSCs were plated at a density of 10,000 cells/well (cell amount was optimized to obtain reliable measurements and minimize variability) and cultured 3 days in GM. The day of the assay, GM was substituted with Seahorse Base XF Base Medium (cat.#102353-100, Seahorse Biosciences) supplemented with Glucose (10 mM) and L-Glutamine (2 mM) (pH = 7.4). Cells were then equilibrated in a non-CO$_2$ incubator (37 °C) for 40–60 min before running the Seahorse XFp Cell Mito Stress Test. The different compounds were serially injected to obtain a final concentration in the wells of: 1 μM Oligo-mycin A (cat.#75351, Sigma), 2.5 μM FCCP (cat.#2920, Sigma), 2 μM Rotenone (cat.#R8875, Sigma), and 2 μM Antimycin A (cat.#A8674, Sigma). After the assay, cells were fixed as previously described and nuclei stained using Hoechst 33342 (cat.#H3570, Invitrogen). ImageJ was used to quantify nuclei number for data normalization.

**Quantification and statistical analysis.** Data are represented as mean ± SEM. The investigators were not blinded to allocation during experiments or to mouse genotypes. Sample size was chosen based on literature and variability observed in previous experience in the laboratory. Comparisons between groups used the Student's $t$ test assuming two-tailed distributions with an alpha level of 0.05. For ChIP studies, comparisons between groups used the Student's $t$ test assuming one-tailed distributions with an alpha level of 0.05. Comparisons between the Pearson's correlation coefficients were performed using one-way analysis of variance (ANOVA). Comparisons between the OCR curves were performed using ordinary two-way ANOVA. Statistical tests were performed using GraphPad Prism 7 or Microsoft Excel for Macintosh.

**Reporting summary.** Further information on research design is available in the Nature Research Reporting Summary linked to this article.

## Data availability

The authors declare that all data supporting the findings of this study are available within the article and its supplementary information files or from the corresponding author upon reasonable request. FASTQ files from the RNA-seq performed on freshly isolated MuSCs from Pax7-CreER;Stat3$^{f/f}$ mice described in this paper have been deposited in the Sequence Read Archive (SRA) database under accession code PRJNA510443. Public MyoD ChIP-seq data from C2C12 myotubes were downloaded from GEO under the code GSM1197185. Raw data files for the MyoD ChIP-seq from IMR90 human cells have been deposited in the SRA database under accession code PRJNA526256. Processed data for the MyoD ChIP-seq from IMR90 human cells have been deposited in GEO under the accession code GSE128527. A reporting summary for this article is available as a Supplementary Information file.

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

## Acknowledgements

This work was supported by the California Institute for Regenerative Medicine (CIRM) training grant TG2-01162 and AFM-Téléthon Postdoctoral fellowship (No. 21084) to D. S.; the US National Institutes of Health (NIH) grants F32 AR070630 to M.J.S.; R01 AR064873, R03 AR063328, and P30 AR061303 to A.S.; R01 AR067731 to G.D.; R01AR056712, R01AR052779, and P30 AR061303 to P.L.P.; Muscular Dystrophy Association grant 382221 to A.S.; SBP Fishman Fund Fellowship to U.E.; Muscular Dystrophy Association, AFM and EPIGEN grants to P.L.P.; and AFM 20568 and the Italian Ministry of Health grant PE-2016-02363049 to L.L. We thank the following people at the SBP Core Facilities for technical support: B. Charbono, D. Sandoval, and A. Vasquez from the Animal Facility; A. Cortez and Y. Altman from the Flow Cytometry Core Facility; L. Boyd from the Cell Imaging Facility; D. Scott from the Cancer Metabolism Facility; and J. Marchica, J. Li, and R. Perera from the Genomics Facility. We would also like to thank the Transgenic Core Facility at UCSD.

## Author contributions

Conceptualization, D.S. and A.S.; methodology, D.S., T.J.C., M.J.S., U.E., C.N., A.D., L.L., and A.S.; formal analysis, D.S.; investigation, D.S., T.J.C., M.J.S., U.E., C.N., and L.L.; writing—original draft, D.S. and A.S.; writing—review and editing, D.S., T.J.C., M.J.S., U. E., C.N., P.L.P., G.D., L.L., and A.S.; supervision, A.S.; funding acquisition, D.S, P.L.P., G. D., L.L., and A.S.

**Additional information**

**Competing interests:** The authors declare no competing interests.

