## [Peer Review File · Nature Communications]

Reviewers' Comments:

Reviewer #1:

Remarks to the Author:

This study investigates the role of Fam3a as a downstream effector of the Stat3 pathway. As more information becomes available on the role of metabolic processes and the commitment and differentiation status of different cell types, this study provides a nice link between a signalling pathway that had previously shown to play an important role in myogenesis, and Fam3a that appears to be regulated by MyoD to modulate cell differentiation. The authors use different strategies and investigate different developmental time points to assess the role that Fam3a plays in the myogenic process. In this way, it provides a link between Stat3 and the downstream differentiation process. The study is therefore interesting, yet in some parts lacking in some depth to appreciate the role of Fam3a.

Comments:

- 1) Fam3a appears to be a target of Stat3 using different criteria. Fig 2e indicates stimulation by IL6 and presence of H3K27Ac. Please indicate the significance values for these histograms; the error bars are quite large and it is not clear if the differences are indeed significant.
- 2) Fig 2f – shRNA shows a decrease by about 50% in expression of Fam3a. Is it not surprising that alterations in ratios of Pax7/MyoD are observed with heterozygous levels? It would be important to assess if part of the population experienced >50% KD and other cells remained WT, and as such giving an average of about 50% reduction in expression of Fam3a.
- 3) Fig 3 – this is a nice experiment showing 2 timepoints for analysis. There are a few interpretation issues that need to be addressed. As this is a full KO, the reduction in muscle may be resulting from non-cell autonomous mechanisms, particularly because Fam3a is affecting metabolism and it is a secreted protein. For example, in Fig 3b the limb looks smaller in the mutant. Is this the case or are these different parts of the limb that are sectioned? Was the muscle area normalized to the relative size of the limb?
- 4) Fig 3c – The image provided shows more Pax7+ cells, but the histogram quantification shows no significant difference in the KO. Please provide representative images. Please also discuss the respective roles of Fam3a in the fetus and P0 – lack of increase in Pax7+ cells in the fetus compared to P0. If Pax3 staining was done on the fetus, would there be an increase given the more important role that Pax3 plays in pre-fetal stages of development? Also, here and in 3d the laminin looks more disrupted in the KO. Is this by chance or is this a phenotype in the Fam3a KO?
- 4) The analysis of non-muscle cells shows that Fam3a is produced by other cell types in the muscle, especially at higher levels in FAPS. To complete the analysis, a comparison between the levels produced by these cell types during growth, homeostasis and injury will enrich the study and provide a better view of how Fam3a contributes to muscle commitment, particularly during the dynamic process of degeneration and regeneration. In this context, do young and old myotubes produce similar levels of Fam3a? To what extent does the myofiber contribute this molecule compared to other cell types?
- 5) Which of these cell states or cell types actually produce the secreted molecule? Since Fam3a is a secreted molecule, in part, according to the present study, how and when this molecule acts and to what extent it is bioavailable to the myogenic cells is important to ascertain. Since the number of myogenic cells is far less than other cell types in the tissue, it appears unlikely that Fam3a produced by mono-nucleated myogenic cells alone plays a major role in the commitment of these cells.
- 6) In a similar vein, since the subcellular localization of Fam3a is different to what was reported in the literature, does this localization change in different phases of myogenic commitment and

differentiation, such as young and older myofibers?

7) The authors show that Fam3a increases basal and maximal respiration by about 20% (OCR). It would be important to know to what extent Fam3a impacts on glycolysis relative to oxidative phosphorylation experimentally. As more data is emerging implicating OxPhos in differentiation, does Fam3a impact both processes or does it favor one over the other. This in the context of the model proposed that Fam3a promotes muscle differentiation. How does this relate to nuclear as opposed to mitochondrial localized Stat3?

Reviewer #2:

Remarks to the Author:

In this article, the authors described that Fam3a is a secreted factor downstream Stat3, fundamental for the myogenic differentiation process. They claim that it acts by inducing mitochondrial respiration.

Despite of the fact that this work describes a new important molecular pathway for myogenic differentiation, it lacks important experiments in order to support their conclusions.

I believe that two functional aspects must be addressed in order to make this work suitable for the publication in Nature Communication.

1. The authors claim that: "Stat3-Fam3a axis promotes muscle cell myogenic lineage progression by inducing mitochondrial respiration".

How does Fam3a promotes mitochondrial respiration? What is the molecular mechanism? What are the targets genes of Fam3a?

They must go deeper in the molecular mechanism. The results of the Seahorse assays just show the final effect but do not clarify the mechanism.

In order to fill full this point, they should check whether Fam3a increases mitochondrial respiration by increasing mitochondrial content (e.g. using mt-GFP construct and quantify it at microscopy). They should check if Fam3a upregulate PGC1alpha. They should check whether Fam3a upregulates the expression of specific mitochondrial subunits of the complexes of the respiratory chain and if there are any of them that are specifically expressed during myogenic differentiation. Moreover, since the formation of SuperComplexes of the respiratory chain has been associated with differentiation in adipocytes¹ and in heart², it will be important to see whether in myogenic differentiation this happen too. I am aware that working with MuSc can be a limitation for such experiments but the authors can try use others cells types and translate the results to MuSC (they claim that Fam3a is a paracrine and autocrine factor and it is also expressed in other cells types. This could be an indication that it maybe has the same role in others cell types). Does Fam3a increase mitochondrial respiration in terminally differentiated muscle cells also?

Moreover, they should calculate the ECAR status of the cells. It can be easily calculated form the Seahorse data already available.

2. If the boost of mitochondrial respiration is an important factor for the myogenic lineage progression, the differentiation should be impaired in mouse models harbouring mutations in some subunits of the mitochondrial respiratory complexes. This will point out unequivocally that the mitochondrial respiration is required for myogenic differentiation. The Cox15sm/sm mouse model³ either Ndusfa4-/- mouse model⁴ could be interesting tools to address this point.

Others important points:

- Fig 1d: what does the red line represent? Explain it in the legend. In the legend of figure 1, the c panel is indicated as d. It must be correct.
- Fig.1e, 4g and S4h: in the graphs, the injections of the drugs must be indicated. It is reported in the material and methods but it is convenient to report it in the graph also.
- In the page 6, the authors say: "we analysed the metabolic properties of MuSCs..." This

statement is too general, especially because they measure only the mitochondrial respiration and they don't mention any other metabolic parameters.

- Fig. 2g and h: how many cells did they count for the quantification?
- Fig 3: in this figure, they show results from Embryos and Pups where Fam3a has been knock out but they didn't show any results that confirm the knock out of Fam3a.
- Do the Pups Fam3a ko show some phenotype? Do they move properly? What about their growth?
- They mention the fig. S3b but it doesn't exist.

• A big part of the work is about the localization of Fam3a capitalized on microscopy technique.

This part present two major problems: (i) one is the expression of a Fam3a with a Myc-Flag tag, (ii) the second is the lack of important controls in the co-localization experiments.

(i) The expression of Fam3a with a tag can modify the localization of the protein. If an antibody against Fam3a exist, I suggest to use it. An interesting experiment, could be isolated mitochondria from muscle and incubate with Fam3a synthesized in vitro and check whether Fam3a localize to mitochondria. Moreover, the sequence of Fam3a should be analysed in order to find/exclude any mitochondrial targeting sequence. These experiments will make stronger the data from co-localization experiments.

(ii) The experiments of co-localization are based on labelling the cellular compartments using specific markers. However, the authors must be sure that the markers label the right compartment in their experimental conditions. I suggest to check these double labelling: GM-130 with TOM20, GM-130 with KDEL, TOM20 with KDEL. If their labelling would be specific, they shouldn't co-localize. Looking at the results of the co-localization experiments, I guess that the purple colour means co-localization (anyway, the colour code must be specified). If I look at the images presented, I can see purple colour also with TOM20 (Fig. S4b) but the authors claim that Fam3a doesn't localized to mitochondria. So, how do they explain it? Please, provide also a quantification of the localization, indicating how many cells are used for it.

On pag.9, about localization of Fam3a, they claim: "its subcellular localization may be context-dependent": what does it mean? What is the context element responsible of the different localization? Please explain better this point.

- In the fig. 4f, the bars show the same difference between unt and Fam3a in both basal and maximal respiration however the authors reported different level of significance. Why it is so?
- Fig 4h: How many cells have been used for the quantification? It will be better to show images more representative of the quantification.

1. Hofmann, A. D. et al. Oxphos supercomplexes as a hallmark of the mitochondrial phenotype of adipogenic differentiated human MSCS. *PLoS One* 7, (2012).
2. Gomez-Velazquez, M. et al. CTCF counter-regulates cardiomyocyte development and maturation programs in the embryonic heart. *PLoS Genet.* 13, 1–25 (2017).
3. Viscomi, C. et al. In vivo correction of COX deficiency by activation of the AMPK/PGC-1 α axis. *Cell Metab.* 14, 80–90 (2011).
4. Calvaruso, M. A. et al. Mitochondrial complex III stabilizes complex I in the absence of NDUFS4 to provide partial activity. *Hum. Mol. Genet.* 21, 115–120 (2012).

Reviewer #3:

Remarks to the Author:

In this manuscript, the authors identified Fam3a as a novel downstream target of Stat3. They report that Fam3a is potentially secreted by myogenic cells. In addition, they show that Stat3 promote mitochondrial respiration during muscle stem cell myogenic lineage progression.

Specific comment:

1. Fig 2e: The authors use IL-6 to stimulate Stat3 activation. Please justify why this cytokine was selected and not others known to stimulate Stat3 (such as LIF or OSM).
2. Fig 2f: Does Fam3a protein level also decrease in Stat3 KO cells?
3. Fig 2g-h: The authors state that Fam3a promotes mitochondrial respiration in vascular smooth muscle cells, neuronal cells and hepatocyte, but they do not address this directly in MuSCs. This could be addressed by assessing the metabolic properties of Fam3a null cells.
4. Fig 2h: Explain why the percentage of myogenin+ cells was calculated in the null cells cultured in growth media and not cultured in the differentiation media? 30% myogenin+ cells seem relatively high for cells cultured in growth media.
5. The authors need to provide more compelling data showing that the synergistic interplay between Stat3 and MyoD on the Fam3a promoter regulates its transcriptional activity in MuSCs. I suggest using luciferase assay with STAT3 +/- MyoD.
6. Do the MyoD null myoblasts also show a reduction in the expression of Fam3a? Do these cells exhibit an abnormal mitochondrial respiration?
7. Fig 3: DAPI staining should be shown. The Pax7 staining seems to be in the center of the fibers in Fig 3c?
8. Fig 4d&e: Again, quantification of myogenin expression should be tested under differentiation media. Please clarify if the experiments in these two panels are done in C2C12 cells or in freshly isolated MuSCs.
9. Fig 4h: Confirmation of the role of Fam3a on muscle regeneration in vivo in the rescue study should be performed by quantifying myogenin+ cells.
10. Most of their in vitro experiments were done in C2C12 cells, I would like to see these experiments to be repeated in primary myoblasts or freshly isolated satellite cells.
11. Figure legends: Typographical error in Fig 1 panel d.

Point by point response to Reviewers

We were pleased to learn that the Reviewers found our work of considerable potential interest. We thank the Reviewers for their insightful comments. We have now performed a substantial amount of new experiments to address all their concerns and we feel they significantly enriched the manuscript. Below are our detailed point-by-point responses.

Reviewer #1:

This study investigates the role of Fam3a as a downstream effector of the Stat3 pathway. As more information becomes available on the role of metabolic processes and the commitment and differentiation status of different cell types, this study provides a nice link between a signalling pathway that had previously shown to play an important role in myogenesis, and Fam3a that appears to be regulated by MyoD to modulate cell differentiation. The authors use different strategies and investigate different developmental time points to assess the role that Fam3a plays in the myogenic process. In this way, it provides a link between Stat3 and the downstream differentiation process. The study is therefore interesting, yet in some parts lacking in some depth to appreciate the role of Fam3a.

Comments:

Point 1 – *Fam3a appears to be a target of Stat3 using different criteria. Fig 2e indicates stimulation by IL6 and presence of H3K27Ac. Please indicate the significance values for these histograms; the error bars are quite large and it is not clear if the differences are indeed significant.*

RE: We apologize for this omission. We have now further performed additional ChIP experiments and indicated statistical analysis of Stat3 binding to the Fam3a promoter (**new Figure 2e**). We observed statistically significant Stat3 enrichment on the Fam3a promoter upon treatment with IL-6 compared to untreated cells ($p=0.037$) and to the IgG control ($p=0.036$). Consistently, H3K27Ac was also significantly enriched on the Fam3a promoter after IL-6 treatment compared to untreated or IgG controls ($p=0.034$ and $p=0.018$, respectively). We also further performed luciferase reporter assays and demonstrate that Stat3 promotes the transcriptional activity of the Fam3a reporter (see also new **Figure 2i** and Response to Reviewer 3, point 5). Together, these data indicate that Fam3a is a direct target of Stat3 in myogenic cells.

Point 2 – *Fig 2f – shRNA shows a decrease by about 50% in expression of Fam3a. Is it not surprising that alterations in ratios of Pax7/MyoD are observed with heterozygous levels? It would be important to assess if part of the population experienced >50% KD and other cells remained WT, and as such giving an average of about 50% reduction in expression of Fam3a.*

RE: We have now assessed whether the levels of Fam3a repression using the shRNA were due to the efficiency of MuSC infection or to the efficiency of shRNA in downregulating the Fam3a gene. To this aim, we used lentiviruses encoding for the shRNA against Fam3a and also for GFP to monitor the percentage of the MuSC population that was infected. Upon infection, we observed ~ 40% downregulation of Fam3a at the population level (**Figure R1**). The infection efficiency was ~ 100% (**Figure R1**), indicating that the repression of Fam3a in our system depended on the efficiency of the shRNA. Moreover, in this condition we observed a significant expansion of the Pax7⁺/MyoD⁺ population and a reduction in the Pax7⁻/MyoD⁺ population (**Figure R1**), consistent with the results already included in the manuscript (see also **Figure 3** in the manuscript). Given that Fam3a is a secreted protein, we believe that the levels of repression we obtained with our shRNAs are sufficient to reduce the amount of Fam3a in the environment (media in the well) and see an impact in the whole MuSC population.

Figure R1. Fam3a repression compromises myogenic lineage progression. (a) Fam3a mRNA levels in MuSCs infected with lentiviruses encoding for shRNA Control (shCt) or shRNA against Fam3a (shFam3a) together with GFP. (b-d) Immunofluorescence analysis and quantification of the efficiency of infection and myogenic markers of MuSCs infected with lentiviruses encoding for shRNA Control (shCt) or shRNA against Fam3a (shFam3a) together with GFP. Cells were cultured in growth conditions for 72 hours (n=3 independent experiments, > 650 cells per condition were quantified). Scale bar 50 μ m. Data represented as mean \pm SEM (Student's t-test; * p<0.05, ** p<0.01).

Point 3 – Fig 3 – this is a nice experiment showing 2 timepoints for analysis. There are a few interpretation issues that need to be addressed. As this is a full KO, the reduction in muscle may be resulting from non-cell autonomous mechanisms, particularly because Fam3a is affecting metabolism and it is a secreted protein. For example, in Fig 3b the limb looks smaller in the mutant. Is this the case or are these different parts of the limb that are sectioned? Was the muscle area normalized to the relative size of the limb?

RE: We agree with the reviewer that we cannot rule out the possibility that the phenotype we observed in the skeletal muscle of the full Fam3a KO mice may also result from cell non-autonomous effects. Indeed, one possibility is that the Fam3a produced in other tissues could affect skeletal muscle development and/or regeneration. Additionally, we have performed comprehensive gene expression analysis in different muscle-resident cell types that suggests that non-myogenic cells also produce Fam3a (see answer to Points 5 and 6). Thus, we do not state in the manuscript that the phenotype of the Fam3a KO mice is due a cell-autonomous effect, but rather we focus our explanation on the functional impact of Fam3a absence on myogenic progenitors independently of the source.

As the reviewer indicated, the limbs from Fam3a KO E15 embryos are smaller than WT embryos (the section analyzed in all the samples corresponded to the same region of the hindlimb, i.e., the part right under the knee). To further characterize our model, we have now normalized the eMyHC area with the diameter of the tibia bone

Figure R2. Effects of Fam3a ablation in bone size and muscle mass in E15 embryos and P0 pups. (a) Tibia diameter and relative muscle area in WT and Fam3a KO E15 embryos (n=5 embryos per group). (b) Tibia diameter and relative muscle area in WT and Fam3a KO P0 pups (n=3-4 pups per group). Data represented as mean \pm SEM (Student's t-test; * p<0.05, ** p<0.01).

(**Figure R2**). The tibia in Fam3a KO embryos was smaller than WT controls, which resulted in no statistical significant differences in eMyHC area after normalization in embryos. However, P0 pups did not display differences in the diameter of the tibia, and the muscle area was significantly smaller even after normalization (**Figure R2**). This data suggests the existence of stage-specific actions of Fam3a during mouse development.

We also assessed skeletal muscle mass in adult (3-month-old) WT and Fam3a KO mice. Fam3a ablation resulted in reduced body weight in adult male mice, together with a significant reduction in the gastrocnemius, quadriceps and heart weights (**Figure R3**). We did not observed significant reduction in the weight of the other tissues analyzed.

Overall, our data indicate that while Fam3a KO mice do not only have a skeletal muscle specific phenotype, Fam3a ablation has a major impact on skeletal muscle mass and growth.

Figure R3. Fam3a KO adult mice display reduced skeletal and heart muscle mass. Body and tissue weights from 3-month-old WT and Fam3a KO male mice (n=5-6 mice per group). Data represented as mean ± SEM (Student's t-test; * p<0.05).

Point 4 – Fig 3c – The image provided shows more Pax7+ cells, but the histogram quantification shows no significant difference in the KO. Please provide representative images. Please also discuss the respective roles of Fam3a in the fetus and P0 – lack of increase in Pax7+ cells in the fetus compared to P0. If Pax3 staining was done on the fetus, would there be an increase given the more important role that Pax3 plays in pre-fetal stages of development? Also, here and in 3d the laminin looks more disrupted in the KO. Is this by chance or is this a phenotype in the Fam3a KO?

RE: We now provide more representative images in **new Figure 4c**. We have also included a brief discussion regarding the role of Fam3a in fetus and P0 pups (page15, lines 25-26; page 16 lines 1-5). Our data indicates that the dynamics of the myogenic progenitors are altered during skeletal muscle development due to Fam3a ablation. There is a reduction in the number of Pax7+ cells from E15 to P0 during healthy development in WT embryos, likely due to the commitment of these cells and contribution to the myofibers. However, Fam3a absence delays this process and results in reduced number of committed progenitors (Myogenin+) at E15 and also increased number of uncommitted MuSCs (Pax7+) at P0 in Fam3aKO mice.

We have also performed Pax3 staining in WT and Fam3aKO embryos at E10.5, when Pax3 expression is abundant (Hutcheson et al., 2009; Relaix et al., 2006; Relaix et al., 2004). We did not observe differences in its expression between WT and Fam3a KO samples (**Figure R4**), suggesting that Fam3a plays a role later in skeletal muscle development.

Finally, we confirm that Fam3a KO pups display more disrupted laminin compared to WT controls. We have now incorporated this aspect into the text (page 10, line 15).

Figure R4. Fam3a ablation did not affect the pattern of expression of Pax3 in the hindlimbs of E10.5 embryos. Pax3 immunofluorescence in the premuscle masses of WT and Fam3a KO E10.5 embryos that will originate the hindlimbs. Scale bar 50 μ m.

Point 5 and 6 – *The analysis of non-muscle cells shows that Fam3a is produced by other cell types in the muscle, especially at higher levels in FAPS. To complete the analysis, a comparison between the levels produced by these cell types during growth, homeostasis and injury will enrich the study and provide a better view of how Fam3a contributes to muscle commitment, particularly during the dynamic process of degeneration and regeneration. In this context, do young and old myotubes produce similar levels of Fam3a? To what extent does the myofiber contribute this molecule compared to other cell types?*

Which of these cell states or cell types actually produce the secreted molecule? Since Fam3a is a secreted molecule, in part, according to the present study, how and when this molecule acts and to what extent it is bioavailable to the myogenic cells is important to ascertain. Since the number of myogenic cells is far less than other cell types in the tissue, it appears unlikely that Fam3a produced by mono-nucleated myogenic cells alone plays a major role in the commitment of these cells.

RE: We thank the reviewer for raising these excellent points. We have now performed a comprehensive analysis of Fam3a expression during skeletal muscle growth, homeostasis and regeneration in MuSCs, FAPs, macrophages (MPs) and myofibers (see **new Figure 5f-g** and **new Supplementary Figure 5**). To this aim, we isolated mononucleated cells (MuSCs, MPs, and FAPs) and intact myofibers from P15 pups and 3-month-old adult mice (see **new Supplementary Figure 5**). Our results show that myofibers express higher levels of Fam3a compared to MuSCs, FAPs and macrophages (see **new Figure 5f**). This data suggests that myofibers are the main producers of Fam3a in skeletal muscle during postnatal growth and adult homeostasis. This aspect is now included in the results and discussion of the manuscript (page 12, lines 14-18; page 16, lines 6-17). Moreover, myofibers isolated from adult skeletal muscle expressed higher levels of Fam3a compared to myofibers isolated from P15 pups, suggesting that Fam3a production increases during myofiber maturation (see **new Figure 5f**).

To assess Fam3a expression during skeletal muscle regeneration, we isolated mononucleated cells from adult male mice (3-month-old) at three different time points: uninjured, 3 days and 7 days after injury. We took into consideration the dynamic changes in the number of the different cell types, and calculated Fam3a mRNA abundance per cell population (see **new Figure 5g**). Furthermore, for the two time-points after injury, we isolated two different populations of MuSCs, $\alpha 7$ -integrin⁺CD34⁺ and $\alpha 7$ -integrin⁺CD34⁻ progenitors, as CD34 expression is reduced in MuSCs upon activation (Sacco et al., 2008; Tierney et al., 2016). 3 days after injury both macrophages and FAPs substantially increased Fam3a expression at the population level (see **new Figure 5g**). 7 days after injury FAPs are the main expressing cells in the tissue, suggesting that FAPs may be a relevant source of Fam3a during skeletal muscle repair (see **new Figure 5g**). We speculate that at later time-points of regeneration, with the formation of the new myofibers and the clearance of FAPs, the new myofibers would become the main source of Fam3a in the regenerating muscle. This aspect is now included in the results and the discussion of the manuscript (page 12, lines 19-24; page 16, lines 6-17).

Point 7 – In a similar vein, since the subcellular localization of Fam3a is different to what was reported in the literature, does this localization change in different phases of myogenic commitment and differentiation, such as young and older myofibers?

RE: We have now analyzed Fam3a localization in C2C12s myotubes overexpressing Fam3a (**Figure R5**). Similar to what we observed in C2C12s myoblasts (see **new Figure 5a-b** and **new Supplementary Figure 4c-e**), exogenous Fam3a was localized in the Golgi and ER (colocalized with GM-130 and KDEL markers). Thus, this data indicates that Fam3a is produced by differentiated myogenic cells, consistent with elevated levels of Fam3a expression we observed in vivo in myofibers.

Figure R5. Exogenous Fam3a localizes in Golgi and ER in C2C12s myotubes. Immuno-fluorescence analysis of C2C12 myogenic cells transfected with a construct to overexpress Fam3a-Myc-Flag and cultured in differentiation media for 5 days. GM-130 is a cis-Golgi marker. KDEL is an ER marker. Pink and yellow colors indicate colocalization. Scale bar 20 μ m.

Point 8 – The authors show that Fam3a increases basal and maximal respiration by about 20% (OCR). It would be important to know to what extent Fam3a impacts on glycolysis relative to oxidative phosphorylation experimentally. As more data is emerging implicating OxPhos in differentiation, does Fam3a impact both processes or does it favor one over the other. This in the context of the model proposed that Fam3a promotes muscle differentiation. How does this relate to nuclear as opposed to mitochondrial localized Stat3?

RE: We thank the reviewer for this suggestion. We have now incorporated the ECAR data into the manuscript as readout of glycolysis (see **new Figures 1g, 6e, Supplementary Figure 3c, Supplementary Figure 6e**). Our results indicate that both Stat3 and Fam3a promote specifically mitochondrial respiration but do not affect glycolysis rate during MuSC lineage progression.

We also now provide evidence reinforcing the role of nuclear Stat3 during MuSC myogenic lineage progression as opposed to mitochondrial Stat3. We have incorporated Venn diagrams representing the transcriptional effects of Stat3 during MuSC activation (**new Figure 1c**). The top 20 differentially enriched pathways during the activation of control MuSCs grouped in three processes: proliferation, protein synthesis and mitochondrial function/biogenesis (**new Figure 1c**). From these three types of processes, only mitochondrial-related pathways were not modulated during the activation in Stat3 KO MuSCs (**new Figure 1c**). Given that these are transcriptional changes, this data suggests that nuclear Stat3 has a role in regulating mitochondrial function during MuSC activation. However, in order to assess a potential role of mitochondrial Stat3 during MuSC activation, we performed immunofluorescence analysis of both total Stat3 and S727 phosphorylated Stat3 in MuSCs cultured in vitro for 3 days. Stat3 S727 phosphorylation enhances its transcriptional activity in the nucleus, and it is also required to induce the translocation of Stat3 into the mitochondria. Both analyses showed

a predominant nuclear localization for Stat3 (**new Supplementary Figure 1c-d**), suggesting that nuclear Stat3 plays a major role in regulating mitochondrial respiration during MuSC activation.

Although we cannot rule out the possibility that low levels of mitochondrial Stat3 can also impact MuSC function, our evidences showing that (1) Significant transcriptional changes in nuclear-encoded genes associated with mitochondria occur upon Stat3 deletion, (2) Predominantly nuclear localization of Stat3 in activated MuSCs, and (3) Identification of a nuclear encoded gene, Fam3a, as a direct Stat3 downstream target promoting mitochondrial respiration and myogenic progression, point to a major role of nuclear Stat3 as regulator of mitochondrial respiration during MuSC activation.

Reviewer #2:

In this article, the authors described that Fam3a is a secreted factor downstream Stat3, fundamental for the myogenic differentiation process. They claim that it acts by inducing mitochondrial respiration.

Despite of the fact that this work describes a new important molecular pathway for myogenic differentiation, it lacks important experiments in order to support their conclusions.

I believe that two functional aspects must be addressed in order to make this work suitable for the publication in Nature Communication.

Point 1 – *The authors claim that: “Stat3-Fam3a axis promotes muscle cell myogenic lineage progression by inducing mitochondrial respiration”.*

How does Fam3a promotes mitochondrial respiration? What is the molecular mechanism? What are the targets genes of Fam3a?

They must go deeper in the molecular mechanism. The results of the Seahorse assays just show the final effect but do not clarify the mechanism.

In order to fill full this point, they should check whether Fam3a increases mitochondrial respiration by increasing mitochondrial content (e.g. using mt-GFP construct and quantify it at microscopy). They should check if Fam3a upregulate PCG1alpha. They should check whether Fam3a upregulates the expression of specific mitochondrial subunits of the complexes of the respiratory chain and if there are any of them that are specifically expressed during myogenic differentiation. Moreover, since the formation of SuperComplexes of the respiratory chain has been associated with differentiation in adipocytes¹ and in heart², it will be important to see whether in myogenic differentiation this happen too. I am aware that working with MuSC can be a limitation for such experiments but the authors can try use others cells types and translate the results to MuSC (they claim that Fam3a is a paracrine and autocrine factor and it is also expressed in other cells types. This could be an indication that it maybe has the same role in others cell types).

RE: We thank the reviewer for these suggestions. We have now performed extensive studies to analyze the potential molecular mechanism by which Fam3a promotes mitochondrial respiration (see **new Supplementary Figure 3**):

1- We assessed whether Fam3a affects mitochondrial content. We isolated MuSCs from WT and Fam3aKO mice, cultured them for 3 days in growth media, and quantified Tomm20's Corrected Total Cell Fluorescence (CTCF). We did not observe significant differences between WT and Fam3aKO MuSCs, indicating that Fam3a ablation does not affect mitochondrial content (see **new Supplementary Figure 3d**).

2- We utilized loss-of-function and gain-of-function approaches to analyze whether Fam3a regulates the expression of Pgc1a as well as different subunits of the mitochondrial electron transport chain (ETC) in myogenic cells. For loss-of-function, we isolated MuSCs from WT and Fam3aKO and cultured them 3 days in growth or differentiation media (see **new Supplementary Figure 3e-f**). Induction of MuSC differentiation increased the expression of Pgc1a and all the subunits of the ETC tested (see **new Supplementary Figure 3e-f**). However, Fam3a genetic ablation did not affect the expression of these genes neither in growth or differentiation conditions (see **new Supplementary Figure 3e-f**). For gain-of-function, we utilized C2C12 transfected to overexpress Fam3a and cultured in growth or differentiation media (see **new Supplementary Figure 3g-j**). Consistent with our previous data, C2C12 differentiation increased the mRNA levels of Pgc1a and the ETC subunits tested (see **new Supplementary Figure 3g-i**). However, Fam3a overexpression did not change the expression of these genes (see **new Supplementary Figure 3g-i**). Additionally, Fam3a overexpression did not alter the expression of the ETC subunits at the protein level in C2C12s myoblasts (see **new Supplementary Figure 3j**).

3- We have analyzed whether Fam3a affects the assembly of the respiratory complexes and the supercomplexes on the mitochondrial inner membrane during myogenic differentiation (Figure R6). We overexpressed Fam3a in C2C12 myoblasts and cultured them in growth conditions (Mb) or in differentiation conditions to induce the formation of myotubes (MT). Next, we isolated mitochondria from myoblasts and myotubes and performed blue-native gel electrophoresis. Consistent with our previous data, we observed increased abundance of Complex I and Complex V in myotubes compared to myoblasts (Figure R6). However, we did not observe any difference between control and Fam3a overexpressing cells. Moreover, we were not able to detect the presence of supercomplexes in C2C12 myoblasts or myotubes. Given that the presence of supercomplexes has been observed in mitochondria isolated from murine skeletal muscle tissue (Cogliati et al., 2016; Ikeda et al., 2013), our data suggests that in an in vitro setting myogenic differentiation is not sufficient to induce the formation of the supercomplexes.

Figure R6. Blue-native electrophoresis analysis of mitochondrial fractions isolated from C2C12 myoblasts and myotubes. C2C12s were transfected with Fam3a or an empty vector and cultured in growth media to obtain myoblasts (Mb) or cultured in differentiation media to induce the formation of myotubes (MT). Mitochondria were isolated from myoblasts and myotubes with or without Fam3a. Next, mitochondrial fractions were resuspended in sample buffer in the presence of 1% DDM (n-dodecyl- β -D-maltoside) and 1% Digitonin. Equal amounts of protein were loaded in Native gels and subjected to electrophoresis in non-denaturing conditions. One gel was stained with Coomassie and the other three were transferred in PVDF membranes and immunostained with total OXPHOS Rodent WB antibody cocktail (abcam Cat #110413 1:1000), anti-mt-Nd3 antibody (abcam Cat #170681 1:1000), or anti-Atp5b antibody (abcam Cat #14730 1:1000).

Overall, the data currently included in the manuscript proves that Fam3a regulates mitochondrial function in MuSCs, as it does in several cell types (Jia et al., 2014; Song et al., 2015; Wang et al., 2014). We performed all the experiments suggested by the reviewer and provided evidence that at the molecular level Fam3a does not promote mitochondrial respiration by regulating mitochondrial content, biogenesis, or expression/assembly of the ETC complexes, suggesting that alternative molecular mechanisms are taking place. However, we do provide the molecular mechanism by which Stat3 regulates MuSC myogenic lineage progression, which is by regulating the expression of genes that promote mitochondrial respiration, including Fam3a. Additionally, we identify a novel biological mechanism by demonstrating that Fam3a is a secreted protein and that treatment with recombinant protein Fam3a is able to rescue the deficits on mitochondrial respiration and MuSC commitment in loss-of-function studies. While further defining the molecular mechanism by which Fam3a regulates mitochondrial respiration is a very interesting question, we believe that is beyond the scope of this manuscript.

Point 2 - Does *Fam3a* increase mitochondrial respiration in terminally differentiated muscle cells also? Moreover, they should calculate the ECAR status of the cells. It can be easily calculated from the Seahorse data already available.

RE: In order to assess whether *Fam3a* increases mitochondrial respiration in terminally differentiated myotubes, we overexpressed *Fam3a* in C2C12 myoblasts, induced them to differentiate into myotubes, and performed the Seahorse XFp Cell Mito Stress Test. *Fam3a* overexpression did not increase mitochondrial respiration in C2C12 differentiated myotubes (**Figure R7**). This data is consistent with our previous findings that addition of recombinant *Fam3a* in control MuSCs does not further increase mitochondrial respiration (see **Supplementary Figure 6c-e**). Overall, our data indicate that *Fam3a* is required for proper mitochondrial respiration, but that addition of *Fam3a* above a certain threshold does not further increase mitochondrial respiration.

Figure R7. *Fam3a* overexpression does not increase mitochondrial respiration in myotubes. Measurement of the Oxygen Consumption Rate (OCR) and the Extracellular Acidification Rate (ECAR) of C2C12s myotubes that overexpress GFP (Ct) or *Fam3a*-Flag-myc and cultured in differentiation conditions for 5 days (n=3 independent experiments). The Cell Mito Stress Test was performed using the Seahorse XFp Analyzer.

Finally, as suggested by the reviewer, we have included ECAR data from all of the Seahorse studies performed in the manuscript (see **new Figures 1g, 6e, Supplementary Figure 3c, Supplementary Figure 6e**). Our results indicate that both *Stat3* and *Fam3a* promote mitochondrial respiration but do not affect glycolysis during MuSC commitment and differentiation.

Point 3 – If the boost of mitochondrial respiration is an important factor for the myogenic lineage progression, the differentiation should be impaired in mouse models harbouring mutations in some subunits of the mitochondrial respiratory complexes. This will point out unequivocally that the mitochondrial respiration is required for myogenic differentiation. The *Cox15sm/sm* mouse model³ either *Ndusfa4*^{-/-} mouse model⁴ could be interesting tools to address this point.

RE: We agree with the Reviewer that this is a relevant point. There is extensive literature showing that Oxphos activity and mitochondrial respiration is increased during myogenic differentiation in myoblasts and MuSCs (Barbieri et al., 2011; Lyons et al., 2004; Remels et al., 2010; Rodgers et al., 2014; Ryall, 2013; Ryall et al., 2015; Shintaku et al., 2016; Theret et al., 2017; Wagatsuma and Sakuma, 2013). However, to our knowledge, a direct role of mitochondrial respiration in actively promoting myogenic differentiation has only been studied in cultured myoblasts (Biswas et al., 1999; Leary et al., 2002; Pawlikowska et al., 2006; Rochard et al., 2000; Wagatsuma and Sakuma, 2013). In these studies the authors treated myoblasts with different inhibitors of mitochondrial respiration (such as antimycin, CCCP, FCCP, myxothiazol, rotenone or oligomycin) and this resulted in impaired myogenic differentiation (Biswas et al., 1999; Leary et al., 2002; Pawlikowska et al., 2006; Rochard et al., 2000; Wagatsuma and Sakuma, 2013). Inhibition of mitochondrial respiration had an impact on the induction of myogenin expression (Pawlikowska et al., 2006; Rochard et al., 2000; Wagatsuma and Sakuma, 2013), which is equivalent to the phenotype we observe in our *Fam3a* loss-of-function studies.

We have now performed new experiments to validate that mitochondrial respiration is also required in MuSCs for myogenic lineage progression. We used two different approaches: pharmacological and genetic inhibition of mitochondrial respiration.

- 1- In the pharmacological approach, we used CCCP. MuSCs were initially cultured in growth media for 48 hours or differentiation media for 24 hours, and then treated with CCCP (or DMSO as vehicle) for additional 24 hours in the corresponding media. In both culture conditions, the percentage of Myogenin positive cells was reduced in CCCP treated MuSCs, indicating that mitochondrial respiration is also required for proper myogenic commitment and differentiation in MuSCs. These new experiments are now included in the manuscript (**new Supplementary Figure 1e-f**).
- 2- In the genetic approach, we took advantage of the Ndufs4 KO mice (Kruse et al., 2008) (**Figure R8**). In these studies, we isolated MuSCs from WT, Het and KO mice at P21 and cultured them in growth or differentiation media. Our data suggests that Ndufs4 ablation compromises MuSC commitment and differentiation in a dose dependent manner (**Figure R8**), further indicating that mitochondrial respiration plays a major role in promoting these processes.

Figure R8. Ndufs4 ablation reduces MuSC differentiation. Immunofluorescence analysis of myogenin expression in MuSCs isolated from WT, Het and Ndufs4 KO mice and cultured in (a) growth media or (b) differentiation media. Scale bars 100 μ m.

Others important points:

- *Fig 1d: what does the red line represent? Explain it in the legend. In the legend of figure 1, the c panel is indicated as d. It must be correct.*

RE: We apologize for the confusion. We have now corrected the figure legend and included that the red line represents p-value=0.05.

- *Fig.1e, 4g and S4h: in the graphs, the injections of the drugs must be indicated. It is reported in the material and methods but it is convenient to report it in the graph also.*

RE: We have now included the injections of the drugs in all the graphs representing the OCR during the Seahorse XFp Cell Mito Stress Test.

- *In the page 6, the authors say: “we analysed the metabolic properties of MuSCs...” This statement is too general, especially because they measure only the mitochondrial respiration and they don’t mention any other metabolic parameters.*

RE: We have now corrected this sentence to refer specifically to mitochondrial respiration (page 6, line 4).

- *Fig. 2g and h: how many cells did they count for the quantification?*

RE: In these studies we quantified between 508 and 335 cells per condition.

- *Fig 3: in this figure, they show results from Embryos and Pups where Fam3a has been knock out but they didn’t show any results that confirm the knock out of Fam3a.*

RE: We performed sequencing analysis to validate our Fam3a KO model (see **Supplementary Figure 2b-c**). Our strategy to generate the Fam3a KO model focused on generating a frame-shift mutation in the exon 2 of the Fam3a gene that not only alters the ORF, but also generates a Stop codon as early as exons 3 or 4. DNA sequencing analysis has been proven to reliably predict the outcome at the expression level of the target gene when using CRISPR/Cas9 zygotic injection (Cunningham et al., 2017; Li et al., 2013; Nakamura et al., 2014), and all the Fam3a KO mice used in the current manuscript displayed a frame-shift mutation. However, we have now also performed qPCR and Western Blot studies to further validate the absence of Fam3a expression in our model (**Figure R9**). Even though we did not necessarily expect changes in the mRNA levels of Fam3a (we are only generating a small frame-shift mutation), we were able to observe a reduction in the Fam3a mRNA levels in Fam3a KO MuSCs compared to WT cells (when cultured in both growth or differentiation media) (**Figure R9**). This data suggests that the frame-shift mutation in this region not only affects the ORF of the gene, but also the stability of the mRNA.

We have also tried different antibodies to detect endogenous Fam3a by Western Blot (the antibodies tested are: Sigma SAB4300968, Sigma SAB1102488, R&D AF2865), however they all exhibited suboptimal specificity. In our hands, the antibody that worked the best was the R&D AF2865. With this antibody we could detect Fam3a overexpression in C2C12 myoblasts, but not the endogenous protein (**Figure R9**). Thus, we decided to use whole-muscle homogenates from WT and Fam3aKO mice as we anticipated that secreted factors would be retained better in whole tissue homogenates than in cells in culture (where it is released into the media). We observed the disappearance of a band of around 25 KDa in the Fam3aKO samples, validating that Fam3a is not expressed in our model (**Figure R9**). However, we also observed the presence of numerous unspecific bands. We believe that the quality of this antibody is not sufficient to include this data as a figure in the manuscript, but we are open to this option if the Reviewer believes it is acceptable.

Figure R9. Fam3a KO model validation. (a) Fam3a expression analysis of MuSCs cultured in Growth Media (GM) or Differentiation Media (DM) for 72 hours (n=3 independent experiments). (b) Western blot analysis of C2C12s transfected with constructs to overexpress GFP or Fam3a. (c) Western blot analysis of whole-muscle homogenates (from gastrocnemius) from WT and Fam3aKO mice. The top image shows the original blots for the Fam3a antibody. The bottom images show the bands corresponding to Gapdh and Fam3a. Data represented as mean ± SEM (Student's t-test; ** p<0.01, **** p<0.0001).

• Do the Pups Fam3a ko show some phenotype? Do they move properly? What about their growth?

RE: To assess whether Fam3a ablation in mice affects their growth, we analyzed body and tissue weights from 3-month-old WT and Fam3a KO male mice (please see also response to Reviewer 1, point 3). Genetic ablation of Fam3a resulted in reduced body weight (**Figure R10**). We also observed a significant reduction in the gastrocnemius, quadriceps and heart weights, but not in other tissues such as liver, epididymal white adipose tissue, spleen, or kidneys (**Figure R10**). Together, this data indicates that Fam3a deletion reduces growth by specifically affecting certain tissues including skeletal muscle.

Figure R10. Fam3a KO adult mice display reduced skeletal and heart muscle mass. Body and tissue weights from 3-month-old WT and Fam3a KO male mice (n=5-6 mice per group). Data represented as mean \pm SEM (Student's t-test; * $p < 0.05$).

We have also assessed whether Fam3a ablation had any effect on the voluntary movement of these mice. To this aim, we individually housed 2-month-old WT and Fam3a KO male mice in cages containing Low-Profile Whireless Running Wheels. Despite the reduced muscle mass, we did not observe any difference in the voluntary movement between WT and Fam3a KO mice (**Figure R11**).

Figure R11. Fam3a ablation does not affect voluntary movement in mice. 2-month-old WT and Fam3a KO male mice were housed with Running Wheels to assess voluntary movement (n=3-4 mice per group). Data represented as mean \pm SEM.

• They mention the fig. S3b but it doesn't exist.

RE: We apologize for the confusion. **New Supplementary Figure 2b-c** shows the sequencing we performed in the Fam3a KO E15 embryos, P0 pups and adult male mice that we used in this manuscript. All the Fam3a KO mice used in this study showed a frame-shift mutation in the exon 2.

• A big part of the work is about the localization of Fam3a capitalized on microscopy technique. This part present two major problems: (i) one is the expression of a Fam3a with a Myc-Flag tag, (ii) the second is the lack of important controls in the co-localization experiments.

(i) The expression of Fam3a with a tag can modify the localization of the protein. If an antibody against Fam3a exist, I suggest to use it. An interesting experiment, could be isolated mitochondria from muscle and incubate with Fam3a synthesized in vitro and check whether Fam3a localize to mitochondria. Moreover, the sequence of Fam3a should be analysed in order to find/exclude any mitochondrial targeting sequence. These experiments will make stronger the data from co-localization experiments.

RE: We agree with the Reviewer that assessing the localization of the endogenous protein would be ideal. Unfortunately, the quality of the commercially available antibodies is not sufficient to detect endogenous Fam3a.

We have now performed a more detailed analysis of the sequence of Fam3a as suggested by the reviewer (see **new Supplementary Figure 4a-b**). We first used the LocTree3 software, which assigns to each protein sequence a single sub-cellular localization (Goldberg et al., 2014). This software predicted that Fam3a is a secreted protein with 84% accuracy. To strengthen this prediction, we have used two additional softwares: TargetP and MitoFates. TargetP is a software that predicts protein localization by analyzing the presence of N-terminal presequences containing mitochondrial targeting peptides (mTP) or secretory signal peptides (SP) (Emanuelsson et al., 2007). TargetP analysis of the Fam3a protein sequence predicted that Fam3a is a secreted protein (see **new Supplementary Figure 4a**). We have also included the prediction of the Citrate Synthase localization as a positive control for a mitochondrial protein. Finally, we have now analyzed Fam3a protein sequence using MitoFates, an improved software to predict the presence of N-terminal mitochondrial targeting signals and their cleavage sites (Fukasawa et al., 2015). Consistent with the previous predictions, Fam3a did not display mitochondrial localization presequence while it was present in the Citrate Synthase (see **new Supplementary Figure 4b**). Overall, the analysis of the Fam3a protein sequence using three different softwares indicated that Fam3a is a secreted protein that lacks a mitochondrial localization signal.

Finally, while we agree with the reviewer that incubation of Fam3a synthesized in vitro with isolated mitochondria is an interesting approach, we were not able to perform such studies due to technical challenges. However, we believe that all the data currently included in the manuscript, i.e. (1) bioinformatics analysis using 3 softwares indicating that Fam3a is secreted and not mitochondrial, (2) colocalization studies in C2C12s myoblasts showing that Fam3a localizes in the Golgi and in the ER upon inhibition of secretion, (3) western blot studies in C2C12s showing that Fam3a accumulates in the cells upon secretion inhibition and that it is also found in the media, and (4) rescue experiments using recombinant Fam3a, demonstrate that Fam3a is a secreted protein and not a mitochondrial protein.

(ii) The experiments of co-localization are based on labelling the cellular compartments using specific markers. However, the authors must be sure that the markers label the right compartment in their experimental conditions. I suggest to check these double labelling: GM-130 with TOM20, GM-130 with KDEL, TOM20 with KDEL. If their labelling would be specific, they shouldn't co-localize. Looking at the results of the co-localization experiments, I guess that the purple colour means co-localization (anyway, the colour code must be specified). If I look at the images presented, I can see purple colour also with TOM20 (Fig. S4b) but the authors claim that Fam3a doesn't localized to mitochondria. So, how do they explain it? Please, provide also a quantification of the localization, indicating how many cells are used for it.

RE: We thank the Reviewer for the suggestion. We have now performed extensive colocalization studies to validate that: (1) the labeling of the different organelles is specific; (2) exogenous Fam3a is localized in the Golgi in basal conditions in C2C12 myoblasts; (3) exogenous Fam3a is localized in the ER upon treatment with Monensin in C2C12s myoblasts. In order to label Golgi, endoplasmic reticulum (ER) and mitochondria, we used the markers GM-130, KDEL and Tomm20, respectively. These markers have been widely used in the literature to label the corresponding cellular compartments (Bampton et al., 2005; di Ronza et al., 2018; Matsuda et al., 2010; Narendra et al., 2008). Additionally, we have now further validated that there is no or minimal colocalization between the markers used by quantifying the Pearson's Correlation Coefficient (PCC) in immunofluorescence studies. Thus, we did not detect any colocalization between Tomm20 and GM-130 ($PCC = -0.024 \pm 0.012$) or between KDEL and GM-130 ($PCC = -0.047 \pm 0.031$) (see **new Supplementary Figure 4c-d**). Similarly, minimal colocalization was detected between Tomm20 and KDEL ($PCC = 0.338 \pm 0.018$) (see **new Supplementary Figure 4c-d**). Next, we analyzed the colocalization between exogenous Fam3a and the different organelles (see **new Figure 5a-b** and **new Supplementary Figure 4d-e**). In basal conditions, no colocalization of Fam3a was

observed with mitochondria (Flag-Tomm20, PCC=0.172±0.022) or endoplasmic reticulum (Myc-KDEL, PCC=0.167±0.028). Instead, we observed significant Fam3a colocalization with the Golgi (Myc-GM130; PCC=0.516±0.025). Moreover, consistent with a secreted protein, we also observed significant colocalization between exogenous Fam3a and the endoplasmic reticulum after inhibition of secretion with Monensin (Myc-KDEL(Monensin), PCC=0.573±0.027). We have now included new images in the manuscript that accurately represent the results of the quantification of colocalization. We have also added the color codes for colocalization in the Figure legends.

Overall, our colocalization studies together with the Western blot analysis and software analysis indicate that Fam3a is a secreted protein.

On pag.9, about localization of Fam3a, they claim: "its subcellular localization may be context-dependent": what does it mean? What is the context element responsible of the different localization? Please explain better this point.

RE: Fam3a was reported to be a mitochondrial protein in liver, neuronal cells and vascular smooth muscle cells (Chen et al., 2017; Jia et al., 2014; Song et al., 2015). However, our data demonstrates that Fam3a is a secreted protein in skeletal muscle. Thus, we decided to remove this sentence from the manuscript and just indicate that Fam3a is a secreted protein and it is not localized in the mitochondria in myogenic cells.

• In the fig. 4f, the bars show the same difference between unt and Fam3a in both basal and maximal respiration however the authors reported different level of significance. Why it is so?

RE: We have now incorporated into the graph the individual values of each measurement (**now Figure 6c**). The basal respiration from Stat3 KO MuSCs treated with Fam3a displayed more variability between samples than the maximal respiration. Thus, this resulted in different levels of significance after running the statistical test.

• Fig 4h: How many cells have been used for the quantification? It will be better to show images more representative of the quantification.

RE: We now provide more representative images of the quantification (see **new Figure 6g**). For this quantification, the range of independent fields utilized was between 8 and 15 fields per sample (depending on the extent of the injury). Also, the number of Pax7 positive cell quantified ranged between 222 and 616 depending on the sample.

Reviewer #3:

In this manuscript, the authors identified Fam3a as a novel downstream target of Stat3. They report that Fam3a is potentially secreted by myogenic cells. In addition, they show that Stat3 promote mitochondrial respiration during muscle stem cell myogenic lineage progression.

Specific comment:

Point 1 – Fig 2e: *The authors use IL-6 to stimulate Stat3 activation. Please justify why this cytokine was selected and not others known to stimulate Stat3 (such as LIF or OSM).*

RE: We decided to use IL-6 based on the work of our laboratory and the extensive literature available from other groups showing that IL-6 promotes myogenic lineage progression and differentiation of C2C12s and MuSCs by activating Stat3 (Baeza-Raja and Munoz-Canoves, 2004; Hoene et al., 2013; Munoz-Canoves et al., 2013; Price et al., 2014; Serrano et al., 2008; Tierney et al., 2014; Wang et al., 2008). In contrast, LIF and Oncostatin M have an inhibitory effect on myogenic differentiation (Alter et al., 2008; Diao et al., 2009; Jo et al., 2005; Sampath et al., 2018), suggesting that other mediators different from Stat3 may play a role in the signaling of these cytokines in myogenic cells. We also selected IL-6 based on its potential as therapeutic target, as it has been previously shown by our laboratory and other groups that IL-6-Stat3 inhibition improves MuSC function in disease contexts such as Duchenne Muscular Dystrophy or aging (Pelosi et al., 2015a; Pelosi et al., 2015b; Price et al., 2014; Tierney et al., 2014; Wada et al., 2017).

Point 2 – Fig 2f: Does Fam3a protein level also decrease in Stat3 KO cells?

RE: We agree with the Reviewer that assessing Fam3a protein levels in Ct and Stat3 KO MuSCs would be a great addition to the manuscript. Unfortunately, the available antibodies do not have sufficient sensitivity to detect endogenous Fam3a in cells (see also **Figure R9**).

Point 3 – Fig 2g-h: The authors state that Fam3a promotes mitochondrial respiration in vascular smooth muscle cells, neuronal cells and hepatocyte, but they do not address this directly in MuSCs. This could be addressed by assessing the metabolic properties of Fam3a null cells.

RE: We thank the Reviewer for this suggestion. We have now analyzed mitochondrial respiration in Fam3a KO MuSCs (**new Supplementary Figure 3a-c**). To this aim, we isolated MuSCs from WT and Fam3a KO mice, cultured them 3 days in growth media, and performed the Seahorse XFp Cell Mito Stress Test. Our results show that Fam3a genetic deletion reduces basal and maximal mitochondrial respiration in MuSCs (**new Supplementary Figure 3a-b**). Consistent with our previous results, Fam3a ablation did not affect glycolysis (assessed as ECAR) in MuSCs (**new Supplementary Figure 3c**).

Point 4 – Fig 2h: Explain why the percentage of myogenin⁺ cells was calculated in the null cells cultured in growth media and not cultured in the differentiation media? 30% myogenin⁺ cells seem relatively high for cells cultured in growth media.

RE: We calculated the percentage of Myogenin⁺ cells in growth media in order to assess the role of Fam3a in the spontaneous commitment of MuSCs. In the experiment in current Figure 3c, MuSC were infected with lentiviruses. In our hands, treatment with different batches of lentivirus preparation can affect spontaneous MuSC differentiation, generating variability in percentage of Myogenin⁺ cells among experiments. However, we agree with the Reviewer that another relevant question is to assess the role of Fam3a in regulating Myogenin expression in differentiation conditions. Thus, we have now quantified the percentage of Myogenin⁺ cells in control and Fam3a knock-down MuSCs (using an shRNA against Fam3a) cultured in differentiation media for 48 hours (**new Supplementary Figure 6a**). Consistent with our previous data, Fam3a repression reduced the percentage of Myogenin⁺ cells when cultured in differentiation media. Moreover, we also performed rescue studies in this context by adding recombinant Fam3a into the media. Similar to what we previously observed in growth conditions (**Figure 6a**), addition of recombinant Fam3a also increased the percentage of myogenin⁺ cells in shFam3a MuSCs in differentiation conditions (**new Supplementary Figure 6a**). Overall, this data indicates that Fam3a plays a relevant role during both commitment and differentiation of MuSCs.

Point 5 – The authors need to provide more compelling data showing that the synergistic interplay between Stat3 and MyoD on the Fam3a promoter regulates its transcriptional activity in MuSCs. I suggest using luciferase assay with STAT3 +/- MyoD.

RE: We agree with the Reviewer that this is a relevant point. We have now performed luciferase reporter assays to validate that Stat3 and MyoD regulate Fam3a expression (see **new Figure 2i**). To this purpose, we cloned the Fam3a promoter into a pGL3 Luciferase plasmid. Next, the Fam3a-Luc and Renilla plasmids were transfected into 293 cells together with plasmids to overexpress Stat3 and/or MyoD. We decided to use 293 cells because C2C12s already express high levels of endogenous MyoD, preventing the assessment of a potential role of MyoD in the reporter assay. Our results show that Stat3 overexpression increased the luciferase activity in the transfected cells compared to controls (see **new Figure 2i**), further indicating that Fam3a is a direct target of Stat3. Interestingly, overexpression of MyoD induced a higher increase in the activity of the reporter (see **new Figure 2i**), validating that MyoD is also a direct regulator of Fam3a. However, we did not observe further increased luciferase activity when transfecting together Stat3 and MyoD. Thus, we have now modified the text accordingly to more accurately represent the new results. Briefly, we removed the sentence stating that Stat3 and MyoD play a synergistic interplay on the Fam3a promoter. Instead, we propose that there is a regulatory network, including both Stat3 and MyoD, that aims at progressively increasing Fam3a expression and oxidative metabolism during myogenic differentiation (page 15, lines 20-23).

Point 6 – Do the MyoD null myoblasts also show a reduction in the expression of Fam3a? Do these cells exhibit an abnormal mitochondrial respiration?

RE: We thank the Reviewer for this suggestion. We have now assessed Fam3a expression in MuSCs isolated from WT and MyoD $-/-$ mice and cultured in growth media for 3 days. Indeed, we observed a strong downregulation of Fam3a in MyoD-ablated cells (see **new Figure 2h**), providing further evidence that MyoD promotes Fam3a expression in MuSCs.

MyoD has been previously shown to regulate mitochondrial function at different levels. Briefly, MyoD promotes oxidative metabolism and mitochondrial respiration in myotubes in vitro and myofibers in vivo by regulating the transcription of several mitochondrial genes (Shintaku et al., 2016). Moreover, MyoD ablation promotes myoblast survival by regulating the expression of different redox and mitochondrial-associated apoptotic genes (Asakura et al., 2007). However, we did not observe significant differences between the mitochondrial respiration of WT and MyoD $-/-$ MuSCs (data not shown), suggesting the existence of potential compensatory mechanisms through development that maintain mitochondrial function in MuSCs with MyoD genetic ablation. This is consistent with previously published data showing that MyoD KO mice do not display major defects in skeletal muscle development due to the existence of compensatory mechanisms (Asakura et al., 2007; Rudnicki et al., 1992; Rudnicki et al., 1993; Yamamoto et al., 2018). Thus, we decided to perform acute MyoD loss-of-function studies. For this purpose, we transfected C2C12 myoblasts with an siRNA control (siCt) or an siRNA against MyoD (siMyoD), and we analyzed mitochondrial respiration by performing the Cell Mito Stress Test (**Figure R12**). We observed that acute MyoD loss-of-function reduced both basal and maximal mitochondrial respiration in C2C12s myoblasts (**Figure R12**), further indicating that MyoD plays a central role in regulating oxidative metabolism in myogenic cells.

Figure R12. Acute MyoD loss-of-function reduces mitochondrial respiration in C2C12 myoblasts. (a) MyoD expression analysis of C2C12 myoblasts transfected with an siRNA control (siCt) or and siRNA against MyoD (siMyoD) and cultured in growth media (n=3 independent experiments). (b) Measurement of the Oxygen Consumption Rate (OCR) control (siCt) and MyoD knock-down (siMyoD) C2C12 myoblasts cultured in growth conditions (n=3 independent experiments). The Cell Mito Stress Test was performed in a Seahorse XFp Analyzer. Data represented as mean \pm SEM (Student's t-test; * p<0.05, ** p<0.01).

Point 7 – Fig 3: DAPI staining should be shown. The Pax7 staining seems to be in the center of the fibers in Fig 3c?

RE: We have now included the nuclei staining in the Figure (see **new Figure 4**). E15 skeletal muscle displays small myofiber size and immature laminin pattern, which makes it difficult to accurately assess the localization of Pax7 positive cells. However, we do not observe any Pax7 positive nuclei inside the myofibers.

Point 8 – Fig 4d&e: Again, quantification of myogenin expression should be tested under differentiation media. Please clarify if the experiments in these two panels are done in C2C12 cells or in freshly isolated MuSCs.

RE: We apologize for the confusion. It is stated in the Figure legend that the experiments were performed in isolated MuSCs and cultured for 3 days in growth media. We have now repeated these experiments using MuSCs cultured in differentiation media (see **new Supplementary Figure 6a-b**). As stated above (point 4),

shRNA-mediated Fam3a knockdown in MuSCs reduced the percentage of Myogenin positive cells when cultured in differentiation media for 48 hours (see **new Supplementary Figure 6a**). Consistent with our previous results, addition of recombinant Fam3a significantly increased the percentage of Myogenin positive cells in shFam3a MuSCs (see **new Supplementary Figure 6a**), indicating that Fam3a plays a relevant role during MuSC differentiation.

We also performed rescue experiments using control and Stat3 KO MuSCs cultured in differentiation media for 48 hours in vitro (see **new Supplementary Figure 6b**). As expected, Stat3 ablation reduced the number of Myogenin positive cells (see **new Supplementary Figure 6b**). Interestingly, addition of recombinant Fam3a was not able to rescue the defects in differentiation on Stat3 ablated MuSCs (see **new Supplementary Figure 6b**), suggesting that other Stat3 downstream targets are also involved in MuSC differentiation.

Point 9 – *Fig 4h: Confirmation of the role of Fam3a on muscle regeneration in vivo in the rescue study should be performed by quantifying myogenin+ cells.*

RE: We have now quantified the number of Myogenin positive cells in the rescue study in vivo. Indeed, we observed that delivery of recombinant Fam3a significantly increased the number of Myogenin positive cells in injured muscles of Pax7-CreER;Stat3f/f mice (see **new Figure 6h**).

Point 10 – *Most of their in vitro experiments were done in C2C12 cells, I would like to see these experiments to be repeated in primary myoblasts or freshly isolated satellite cells.*

RE: We apologize to the Reviewer for the confusion. Most of the in vitro experiments currently present in the manuscript have been performed using MuSCs.

The in vitro experiments performed in MuSCs include:

- Figure 1e-g (Seahorse in Stat3 KO MuSCs)
- Figure 2d (Fam3a and Stat3 mRNA levels in growth and differentiation conditions)
- Figure 2h (Fam3a mRNA levels in MyoD -/- MuSCs)
- Figure 3a-d (characterization of role of Fam3a in vitro using shRNA)
- Figure 6a (rescue with recombinant Fam3a in shFam3a MuSCs in GM)
- Figure 6b (rescue with recombinant Fam3a in Stat3 KO MuSCs in GM)
- Figure 6c-e (Seahorse in Stat3 KO MuSCs with recombinant Fam3a treatment)
- Supplementary Figure 1c (total Stat3 localization in activated MuSCs in vitro)
- Supplementary Figure 1d (S727 pStat3 localization in activated MuSCs in vitro)
- Supplementary Figure 1e (CCCP treatment on MuSCs grown in GM)
- Supplementary Figure 1f (CCCP treatment on MuSCs grown in DM)
- Supplementary Figure 2a (quantification of proliferation in shFam3a MuSCs)
- Supplementary Figure 3a-c (Seahorse in Fam3aKO MuSCs)
- Supplementary Figure 3d (mitochondrial content in Fam3aKO MuSCs)
- Supplementary Figure 3e-f (qPCR analysis of Pgc1a and ETC genes in Fam3aKO MuSCs)
- Supplementary Figure 6a (rescue with recombinant Fam3a in shFam3a MuSCs in DM)
- Supplementary Figure 6b (rescue with recombinant Fam3a in Stat3 KO MuSCs in DM)
- Supplementary Figure 6c-e (Seahorse in Ct MuSCs with recombinant Fam3a treatment)

C2C12s were only used in conditions when absolutely necessary, i.e., overexpression studies or experiments that required large amount of material. Thus, the experiments performed with C2C12s are limited to:

- Figure 2e (Stat3 ChIP on Fam3a promoter)
- Figure 5a-b (Fam3a colocalization studies)
- Figure 5c-e (WB validation of Fam3a secretion)
- Supplementary Figure 3g-j (expression of Pgc1a and ETC subunits in Fam3a overexpression)
- Supplementary Figure 4c-e (Fam3a colocalization studies)

Point 11 – *Figure legends: Typographical error in Fig 1 panel d.*

RE: We thank the Reviewer for pointing this out. We have corrected this error.

References

- Alter, J., Rozentzweig, D., and Bengal, E. (2008). Inhibition of myoblast differentiation by tumor necrosis factor alpha is mediated by c-Jun N-terminal kinase 1 and leukemia inhibitory factor. *J Biol Chem* 283, 23224-23234.
- Asakura, A., Hirai, H., Kablar, B., Morita, S., Ishibashi, J., Piras, B.A., Christ, A.J., Verma, M., Vineretsky, K.A., and Rudnicki, M.A. (2007). Increased survival of muscle stem cells lacking the MyoD gene after transplantation into regenerating skeletal muscle. *Proc Natl Acad Sci U S A* 104, 16552-16557.
- Baeza-Raja, B., and Munoz-Canoves, P. (2004). p38 MAPK-induced nuclear factor-kappaB activity is required for skeletal muscle differentiation: role of interleukin-6. *Mol Biol Cell* 15, 2013-2026.
- Bampton, E.T., Goemans, C.G., Niranjan, D., Mizushima, N., and Tolkovsky, A.M. (2005). The dynamics of autophagy visualized in live cells: from autophagosome formation to fusion with endo/lysosomes. *Autophagy* 1, 23-36.
- Barbieri, E., Battistelli, M., Casadei, L., Vallorani, L., Piccoli, G., Guescini, M., Giocchini, A.M., Polidori, E., Zeppa, S., Ceccaroli, P., *et al.* (2011). Morphofunctional and Biochemical Approaches for Studying Mitochondrial Changes during Myoblasts Differentiation. *J Aging Res* 2011, 845379.
- Biswas, G., Adebajo, O.A., Freedman, B.D., Anandatheerthavarada, H.K., Vijayasathy, C., Zaidi, M., Kotlikoff, M., and Avadhani, N.G. (1999). Retrograde Ca²⁺ signaling in C2C12 skeletal myocytes in response to mitochondrial genetic and metabolic stress: a novel mode of inter-organelle crosstalk. *EMBO J* 18, 522-533.
- Chen, Z., Wang, J., Yang, W., Chen, J., Meng, Y., Geng, B., Cui, Q., and Yang, J. (2017). FAM3A mediates PPARgamma's protection in liver ischemia-reperfusion injury by activating Akt survival pathway and repressing inflammation and oxidative stress. *Oncotarget* 8, 49882-49896.
- Cogliati, S., Calvo, E., Loureiro, M., Guaras, A.M., Nieto-Arellano, R., Garcia-Poyatos, C., Ezkurdia, I., Mercader, N., Vazquez, J., and Enriquez, J.A. (2016). Mechanism of super-assembly of respiratory complexes III and IV. *Nature* 539, 579-582.
- Cunningham, T.J., Yu, M.S., McKeithan, W.L., Spiering, S., Carrette, F., Huang, C.T., Bushway, P.J., Tierney, M., Albin, S., Giacca, M., *et al.* (2017). Id genes are essential for early heart formation. *Genes Dev.*
- di Ronza, A., Bajaj, L., Sharma, J., Sanagasetti, D., Lotfi, P., Adamski, C.J., Collette, J., Palmieri, M., Amawi, A., Popp, L., *et al.* (2018). CLN8 is an endoplasmic reticulum cargo receptor that regulates lysosome biogenesis. *Nat Cell Biol.*
- Diao, Y., Wang, X., and Wu, Z. (2009). SOCS1, SOCS3, and PIAS1 promote myogenic differentiation by inhibiting the leukemia inhibitory factor-induced JAK1/STAT1/STAT3 pathway. *Mol Cell Biol* 29, 5084-5093.
- Emanuelsson, O., Brunak, S., von Heijne, G., and Nielsen, H. (2007). Locating proteins in the cell using TargetP, SignalP and related tools. *Nat Protoc* 2, 953-971.
- Fukasawa, Y., Tsuji, J., Fu, S.C., Tomii, K., Horton, P., and Imai, K. (2015). MitoFates: improved prediction of mitochondrial targeting sequences and their cleavage sites. *Mol Cell Proteomics* 14, 1113-1126.
- Goldberg, T., Hecht, M., Hamp, T., Karl, T., Yachdav, G., Ahmed, N., Altermann, U., Angerer, P., Ansorge, S., Balasz, K., *et al.* (2014). LocTree3 prediction of localization. *Nucleic Acids Res* 42, W350-355.

Hoene, M., Runge, H., Haring, H.U., Schleicher, E.D., and Weigert, C. (2013). Interleukin-6 promotes myogenic differentiation of mouse skeletal muscle cells: role of the STAT3 pathway. *Am J Physiol Cell Physiol* 304, C128-136.

Hutcheson, D.A., Zhao, J., Merrell, A., Haldar, M., and Kardon, G. (2009). Embryonic and fetal limb myogenic cells are derived from developmentally distinct progenitors and have different requirements for beta-catenin. *Genes Dev* 23, 997-1013.

Ikeda, K., Shiba, S., Horie-Inoue, K., Shimokata, K., and Inoue, S. (2013). A stabilizing factor for mitochondrial respiratory supercomplex assembly regulates energy metabolism in muscle. *Nat Commun* 4, 2147.

Jia, S., Chen, Z., Li, J., Chi, Y., Wang, J., Li, S., Luo, Y., Geng, B., Wang, C., Cui, Q., *et al.* (2014). FAM3A promotes vascular smooth muscle cell proliferation and migration and exacerbates neointima formation in rat artery after balloon injury. *J Mol Cell Cardiol* 74, 173-182.

Jo, C., Kim, H., Jo, I., Choi, I., Jung, S.C., Kim, J., Kim, S.S., and Jo, S.A. (2005). Leukemia inhibitory factor blocks early differentiation of skeletal muscle cells by activating ERK. *Biochim Biophys Acta* 1743, 187-197.

Kruse, S.E., Watt, W.C., Marcinek, D.J., Kapur, R.P., Schenkman, K.A., and Palmiter, R.D. (2008). Mice with mitochondrial complex I deficiency develop a fatal encephalomyopathy. *Cell Metab* 7, 312-320.

Leary, S.C., Hill, B.C., Lyons, C.N., Carlson, C.G., Michaud, D., Kraft, C.S., Ko, K., Glerum, D.M., and Moyes, C.D. (2002). Chronic treatment with azide in situ leads to an irreversible loss of cytochrome c oxidase activity via holoenzyme dissociation. *J Biol Chem* 277, 11321-11328.

Li, D., Qiu, Z., Shao, Y., Chen, Y., Guan, Y., Liu, M., Li, Y., Gao, N., Wang, L., Lu, X., *et al.* (2013). Heritable gene targeting in the mouse and rat using a CRISPR-Cas system. *Nat Biotechnol* 31, 681-683.

Lyons, C.N., Leary, S.C., and Moyes, C.D. (2004). Bioenergetic remodeling during cellular differentiation: changes in cytochrome c oxidase regulation do not affect the metabolic phenotype. *Biochem Cell Biol* 82, 391-399.

Matsuda, N., Sato, S., Shiba, K., Okatsu, K., Saisho, K., Gautier, C.A., Sou, Y.S., Saiki, S., Kawajiri, S., Sato, F., *et al.* (2010). PINK1 stabilized by mitochondrial depolarization recruits Parkin to damaged mitochondria and activates latent Parkin for mitophagy. *J Cell Biol* 189, 211-221.

Munoz-Canoves, P., Scheele, C., Pedersen, B.K., and Serrano, A.L. (2013). Interleukin-6 myokine signaling in skeletal muscle: a double-edged sword? *FEBS J* 280, 4131-4148.

Nakamura, K., Fujii, W., Tsuboi, M., Tanihata, J., Teramoto, N., Takeuchi, S., Naito, K., Yamanouchi, K., and Nishihara, M. (2014). Generation of muscular dystrophy model rats with a CRISPR/Cas system. *Sci Rep* 4, 5635.

Narendra, D., Tanaka, A., Suen, D.F., and Youle, R.J. (2008). Parkin is recruited selectively to impaired mitochondria and promotes their autophagy. *J Cell Biol* 183, 795-803.

Pawlikowska, P., Gajkowska, B., Hocquette, J.F., and Orzechowski, A. (2006). Not only insulin stimulates mitochondriogenesis in muscle cells, but mitochondria are also essential for insulin-mediated myogenesis. *Cell Prolif* 39, 127-145.

Pelosi, L., Berardinelli, M.G., De Pasquale, L., Nicoletti, C., D'Amico, A., Carvello, F., Moneta, G.M., Catizone, A., Bertini, E., De Benedetti, F., *et al.* (2015a). Functional and Morphological Improvement of Dystrophic Muscle by Interleukin 6 Receptor Blockade. *EBioMedicine* 2, 285-293.

Pelosi, L., Berardinelli, M.G., Forcina, L., Spelta, E., Rizzuto, E., Nicoletti, C., Camilli, C., Testa, E., Catizone, A., De Benedetti, F., *et al.* (2015b). Increased levels of interleukin-6 exacerbate the dystrophic phenotype in mdx mice. *Hum Mol Genet* 24, 6041-6053.

Price, F.D., von Maltzahn, J., Bentzinger, C.F., Dumont, N.A., Yin, H., Chang, N.C., Wilson, D.H., Frenette, J., and Rudnicki, M.A. (2014). Inhibition of JAK-STAT signaling stimulates adult satellite cell function. *Nat Med* 20, 1174-1181.

Relaix, F., Montarras, D., Zaffran, S., Gayraud-Morel, B., Rocancourt, D., Tajbakhsh, S., Mansouri, A., Cumano, A., and Buckingham, M. (2006). Pax3 and Pax7 have distinct and overlapping functions in adult muscle progenitor cells. *J Cell Biol* 172, 91-102.

Relaix, F., Rocancourt, D., Mansouri, A., and Buckingham, M. (2004). Divergent functions of murine Pax3 and Pax7 in limb muscle development. *Genes Dev* 18, 1088-1105.

Remels, A.H., Langen, R.C., Schrauwen, P., Schaart, G., Schols, A.M., and Gosker, H.R. (2010). Regulation of mitochondrial biogenesis during myogenesis. *Mol Cell Endocrinol* 315, 113-120.

Rochard, P., Rodier, A., Casas, F., Cassar-Malek, I., Marchal-Victorion, S., Daury, L., Wrutniak, C., and Cabello, G. (2000). Mitochondrial activity is involved in the regulation of myoblast differentiation through myogenin expression and activity of myogenic factors. *J Biol Chem* 275, 2733-2744.

Rodgers, J.T., King, K.Y., Brett, J.O., Cromie, M.J., Charville, G.W., Maguire, K.K., Brunson, C., Mastey, N., Liu, L., Tsai, C.R., *et al.* (2014). mTORC1 controls the adaptive transition of quiescent stem cells from G0 to G(Alert). *Nature* 510, 393-396.

Rudnicki, M.A., Braun, T., Hinuma, S., and Jaenisch, R. (1992). Inactivation of MyoD in mice leads to up-regulation of the myogenic HLH gene Myf-5 and results in apparently normal muscle development. *Cell* 71, 383-390.

Rudnicki, M.A., Schnegelsberg, P.N., Stead, R.H., Braun, T., Arnold, H.H., and Jaenisch, R. (1993). MyoD or Myf-5 is required for the formation of skeletal muscle. *Cell* 75, 1351-1359.

Ryall, J.G. (2013). Metabolic reprogramming as a novel regulator of skeletal muscle development and regeneration. *FEBS J* 280, 4004-4013.

Ryall, J.G., Cliff, T., Dalton, S., and Sartorelli, V. (2015). Metabolic Reprogramming of Stem Cell Epigenetics. *Cell Stem Cell* 17, 651-662.

Sacco, A., Doyonnas, R., Kraft, P., Vitorovic, S., and Blau, H.M. (2008). Self-renewal and expansion of single transplanted muscle stem cells. *Nature* 456, 502-506.

Sampath, S.C., Sampath, S.C., Ho, A.T.V., Corbel, S.Y., Millstone, J.D., Lamb, J., Walker, J., Kinzel, B., Schmedt, C., and Blau, H.M. (2018). Induction of muscle stem cell quiescence by the secreted niche factor Oncostatin M. *Nat Commun* 9, 1531.

Serrano, A.L., Baeza-Raja, B., Perdiguero, E., Jardi, M., and Munoz-Canoves, P. (2008). Interleukin-6 is an essential regulator of satellite cell-mediated skeletal muscle hypertrophy. *Cell Metab* 7, 33-44.

Shintaku, J., Peterson, J.M., Talbert, E.E., Gu, J.M., Ladner, K.J., Williams, D.R., Mousavi, K., Wang, R., Sartorelli, V., and Guttridge, D.C. (2016). MyoD Regulates Skeletal Muscle Oxidative Metabolism Cooperatively with Alternative NF-kappaB. *Cell Rep* 17, 514-526.

Song, Q., Gou, W.L., and Zhang, R. (2015). FAM3A Protects HT22 Cells Against Hydrogen Peroxide-Induced Oxidative Stress Through Activation of PI3K/Akt but not MEK/ERK Pathway. *Cell Physiol Biochem* 37, 1431-1441.

Theret, M., Gsaier, L., Schaffer, B., Juban, G., Ben Larbi, S., Weiss-Gayet, M., Bultot, L., Collodet, C., Foretz, M., Desplanches, D., *et al.* (2017). AMPKalpha1-LDH pathway regulates muscle stem cell self-renewal by controlling metabolic homeostasis. *EMBO J* 36, 1946-1962.

Tierney, M.T., Aydogdu, T., Sala, D., Malecova, B., Gatto, S., Puri, P.L., Latella, L., and Sacco, A. (2014). STAT3 signaling controls satellite cell expansion and skeletal muscle repair. *Nat Med* 20, 1182-1186.

Tierney, M.T., Gromova, A., Sesillo, F.B., Sala, D., Spenle, C., Orend, G., and Sacco, A. (2016). Autonomous Extracellular Matrix Remodeling Controls a Progressive Adaptation in Muscle Stem Cell Regenerative Capacity during Development. *Cell Rep* 14, 1940-1952.

Wada, E., Tanihata, J., Iwamura, A., Takeda, S., Hayashi, Y.K., and Matsuda, R. (2017). Treatment with the anti-IL-6 receptor antibody attenuates muscular dystrophy via promoting skeletal muscle regeneration in dystrophin-/utrophin-deficient mice. *Skelet Muscle* 7, 23.

Wagatsuma, A., and Sakuma, K. (2013). Mitochondria as a potential regulator of myogenesis. *ScientificWorldJournal* 2013, 593267.

Wang, C., Chi, Y., Li, J., Miao, Y., Li, S., Su, W., Jia, S., Chen, Z., Du, S., Zhang, X., *et al.* (2014). FAM3A activates PI3K p110alpha/Akt signaling to ameliorate hepatic gluconeogenesis and lipogenesis. *Hepatology* 59, 1779-1790.

Wang, K., Wang, C., Xiao, F., Wang, H., and Wu, Z. (2008). JAK2/STAT2/STAT3 are required for myogenic differentiation. *J Biol Chem* 283, 34029-34036.

Yamamoto, M., Legendre, N.P., Biswas, A.A., Lawton, A., Yamamoto, S., Tajbakhsh, S., Kardon, G., and Goldhamer, D.J. (2018). Loss of MyoD and Myf5 in Skeletal Muscle Stem Cells Results in Altered Myogenic Programming and Failed Regeneration. *Stem Cell Reports* 10, 956-969.

Reviewers' Comments:

Reviewer #1:

Remarks to the Author:

The authors have done numerous validation experiments and consolidated the study, and have answered the questions raised in an exemplary manner.

For scientific interest, I would encourage the authors to compare their shRNA results (former Fig 2F, now Fig. 3) with the phenotype of Fam3a +/- mice. It is interesting and perhaps surprising that with an shRNA that does not fully decrease expression, a phenotype is observed. Although not unusual, consolidating this finding would be reassuring, also in the context of the cell-non-autonomous action of the gene. This is not critical for publication, but it would be nice to see done at some point.

Reviewer #2:

Remarks to the Author:

Sala and co-authors did a great effort to improve the quality of their manuscript. Indeed, they provided lot of experiments aimed to clarify the questions of the reviewers. Overall, the new version of the manuscript is more detailed and deepens the study of the role of STAT3-Fam3 axis.

In details:

- Point 2: They completely fulfil this point providing the ECAR values also. These results sustain the idea that Fam3 is critical for the MuSC lineage progression only.
- Point 3: I really appreciated the use of two different approaches to validate that mitochondrial respiration is required for MuSC myogenic lineage progression.
- Statement in page 6, line 4 is now more appropriate.
- They did a great job to validate the Fam3a KO model. I appreciated the results of the PCR together with the western-blot. I understand the issue of the antibody specificity and I agree with the authors that the quality of the blot, even though the disappearance of the band is visible, is not sufficient and no needed for publication.
- The bioinformatics analysis of Fam3a sequence and the experiments of co-localization (now improved) definitively demonstrated that Fam3a is not a mitochondrial protein and it doesn't localize to mitochondria.
- The characterization of the Fam3a KO mice is very interesting (body weight, muscle and organs size) and it reveals the presence of a muscular phenotype. I believe that could be an important point that could be interesting for the reader. I suggest them to move these results to supplementary.
- Point 1: the authors did a great effort to describe the molecular mechanism underlying the increase of mitochondrial respiration Fam3a-dependent. However, the experimental conditions of the Bluentaive are incorrect. They used 1% DDM together with 1% Digitonin. DDM is a stronger detergent than Digitonin that disrupts Supercomplexes. For that reason, they could not see them. The standard concentration should be 4% of Digitonin. With this result, they cannot exclude that Fam3a doesn't increase SCs assembly. I suggest to repeat the bluentaive to confirm this aspect.

From the results of the manuscript, it is possible to conclude that Fam3a doesn't increase mitochondrial respiration by modifying mitogenesis, mitochondrial subunits expression while SCs formation is still a doubt. Moreover, Fam3a doesn't localize into mitochondria suggesting that, as clearly stated in the response to Point 1, "at the molecular level Fam3a does not promote mitochondrial respiration by regulating mitochondrial content, biogenesis, or expression/assembly of the ETC complexes, suggesting that alternative molecular mechanisms are taking place". I think that this is an important statement and for the sake of completeness, I suggest to report this clear and precise statement in the main text of the paragraph named: "Fam3a is required for skeletal muscle development in vivo".

Excluding direct effects on mitochondrial component, it seems that Stat3-Fam3a axis could boost mitochondrial respiration by the activation of AKT signalling either Ca²⁺ signalling as commented in the discussion.

In conclusion, I consider the work suitable for the publication on Nature Communication for having described a new function for the Stat3-Fam3a axis and for the discovery of Fam3a as new Stat3 target. The description of the increase of mitochondrial respiration is very relevant but it is necessary to state better that the molecular mechanism underlying this event is still unknown in order to give a clear message to the reader, opening new research possibilities.

Reviewer #3:

Remarks to the Author:

I am satisfied by author's response to my comments.

Minor comments:

1. The authors show that Fam3a mRNA expression level is substantially increased in FAP during skeletal muscle repair. How does this relate to the expression of Stat3 in muscle residing cells (FAPs, MPs)? Does Stat3 regulate the expression of Fam3a in these cells.

2. Fig6h: The y-axis scale unit is too large. Can you please lower the highest value, so that the distinctions between the data sets are more visible.

Point by point response to Reviewers

We would like to thank the Reviewers for their insightful and positive comments on our manuscript. We have now performed new experiments and updated the text/figures to address their final concerns. Below are our detailed point-by-point responses.

Reviewer #1:

The authors have done numerous validations experiments and consolidated the study, and have answered the questions raised in an exemplary manner.

RE: Thank you very much.

For scientific interest, I would encourage the authors to compare their shRNA results (former Fig 2F, now Fig. 3) with the phenotype of Fam3a +/- mice. It is interesting and perhaps surprising that with an shRNA that does not fully decrease expression, a phenotype is observed. Although not unusual, consolidating this finding would be reassuring, also in the context of the cell-non-autonomous action of the gene. This is not critical for publication, but it would be nice to see done at some point.

RE: We thank the reviewer for this suggestion. We have now incorporated in the discussion a sentence stating that partial loss-of-function of Fam3a is sufficient to impair MuSC myogenic commitment, suggesting that there is a requirement of a minimum level of Fam3a to ensure proper myogenic lineage progression (page 16, lines 15-17).

Reviewer #2:

Sala and co-authors did a great effort to improve the quality of their manuscript. Indeed, they provided lot of experiments aimed to clarify the questions of the reviewers. Overall, the new version of the manuscript is more detailed and deepens the study of the role of STAT3-Fam3 axis.

RE: Thank you very much.

In details:

- Point 2: They completely fulfil this point providing the ECAR values also. These results sustain the idea that Fam3 is critical for the MuSC lineage progression only.*
- Point 3: I really appreciated the use of two different approaches to validate that mitochondrial respiration is required for MuSC myogenic lineage progression.*
- Statement in page 6, line 4 is now more appropriate.*
- They did a great job to validate the Fam3a KO model. I appreciated the results of the PCR together with the western-blot. I understand the issue of the antibody specificity and I agree with the authors that the quality of the blot, even though the disappearance of the band is visible, is not sufficient and no needed for publication.*
- The bioinformatics analysis of Fam3a sequence and the experiments of co-localization (now improved) definitively demonstrated that Fam3a is not a mitochondrial protein and it doesn't localize to mitochondria.*

RE: Thank you.

- The characterization of the Fam3a KO mice is very interesting (body weight, muscle and organs size) and it reveals the presence of a muscular phenotype. I believe that could be an important point that could be interesting for the reader. I suggest them to move these results to supplementary.

RE: We agree with the Reviewer. We have now incorporated this data as Supplementary Figure 4 and described it in the text (page 10, lines 20-26), as suggested.

- Point 1: the authors did a great effort to describe the molecular mechanism underlying the increase of mitochondrial respiration Fam3a-dependent. However, the experimental conditions of the Bluenative are incorrect. They used 1% DDM together with 1% Digitonin. DDM is a stronger detergent than Digitonin that disrupts Supercomplexes. For that reason, they could not see them. The standard concentration should be 4% of Digitonin. With this result, they cannot exclude that Fam3a doesn't increase SCs assembly. I suggest to repeat the bluenative to confirm this aspect.

RE: We thank the reviewer for this excellent technical suggestion. We have now repeated this experiment using only Digitonin to homogenize the mitochondrial fractions, and we were able to detect the presence of the Supercomplexes. Our extraction kit did not allow us to use 4% Digitonin, so we used 4g/g detergent/protein ratio as reported on other studies (Jha et al., 2016). Our results show that Fam3a overexpression did not affect the content of the supercomplexes in both myoblasts and myotubes compared to control cells, indicating that Fam3a does not affect the assembly of the supercomplexes in myogenic cells.

Figure R1. Blue-native electrophoresis analysis of mitochondrial fractions isolated from C2C12 myoblasts and myotubes. C2C12s were transfected with Fam3a or an empty vector and cultured in growth media to obtain myoblasts (Mb) or cultured in differentiation media to induce the formation of myotubes (MT). Mitochondria were isolated by resuspending cells in 2 ml of homogenizing buffer (10mM Tris-HCl, 0,25M Sucrose, 10mM HEPES, 1mM EDTA) and homogenized with a motor-driven glass-Teflon mortar for 30 strokes. The homogenate was then centrifuged at 600 g to pellet nuclei. Next, supernatants were spun down at 20.000 g for 30' to pellet mitochondria. Mitochondrial fractions were resuspended in sample buffer in the presence of 4g/g Digitonin/protein ratio (Native PAGE Sample prep Kit Cat # BN2008 Invitrogen). Equal amounts of protein were loaded in Native gels (Native PAGE Cat # BN1002BOX Invitrogen) and subjected to electrophoresis in non-denaturing conditions at 150 mA for 2 hrs. One gel was stained with Coomassie and the other was transferred in PVDF membrane and immunostained with total OXPHOS Rodent WB antibody cocktail (Abcam Cat. #110413 1:1000).

From the results of the manuscript, it is possible to conclude that Fam3a doesn't increase mitochondrial respiration by modifying mitogenesis, mitochondrial subunits expression while SCs formation is still a doubt. Moreover, Fam3a doesn't localize into mitochondria suggesting that, as clearly stated in the response to Point 1, "at the molecular level Fam3a does not promote mitochondrial respiration by regulating mitochondrial content, biogenesis, or expression/assembly of the ETC complexes, suggesting that alternative molecular mechanisms are taking place". I think that this is an important statement and for the sake of completeness, I suggest to report this clear and precise statement in the main text of the paragraph named: "Fam3a is required for skeletal muscle development in vivo".

RE: We agree with the Reviewer. We have now included this statement in the manuscript (page 11, lines 10-13).

Excluding direct effects on mitochondrial component, it seems that Stat3-Fam3a axis could boost mitochondrial respiration by the activation of AKT signalling either Ca²⁺ signalling as commented in the discussion.

In conclusion, I consider the work suitable for the publication on Nature Communication for having described a new function for the Stat3-Fam3a axis and for the discovery of Fam3a as new Stat3 target. The description of the increase of mitochondrial respiration is very relevant but it is necessary to state better that the molecular mechanism underlying this event is still unknown in order to give a clear message to the reader, opening new research possibilities.

RE: We have now clearly stated in the discussion that the mechanism by which Fam3a induces mitochondrial respiration in myogenic cells is currently unknown (page 17, lines 12-15).

Reviewer #3:

I am satisfied by author's response to my comments.

RE: Thank you.

Minor comments:

1. The authors show that Fam3a mRNA expression level is substantially increased in FAP during skeletal muscle repair. How does this relate to the expression of Stat3 in muscle residing cells (FAPs, MPs)? Does Stat3 regulate the expression of Fam3a in these cells.

RE: We have now analyzed Stat3 mRNA levels in MPs and FAPs during skeletal muscle repair. MPs showed a significant increase in Stat3 mRNA levels 3dpi, which correlated with the peak of Fam3a mRNA levels previously observed in these cells (Figure 5g). Regarding FAPs, we did not observe significant changes in Stat3 mRNA levels during the injury process. However, given the activity of the Stat3 signaling pathway is highly regulated by posttranslational modifications (Sala and Sacco, 2016), we cannot rule out the possibility that Fam3a expression is also regulated by Stat3 in these other cells types. Future studies should comprehensively assess this point.

2. Fig6h: The y-axis scale unit is too large. Can you please lower the highest value, so that the distinctions between the data sets are more visible.

RE: We have now lowered the scale of Fig. 6h graph.

References

- Jha, P., Wang, X., and Auwerx, J. (2016). Analysis of Mitochondrial Respiratory Chain Supercomplexes Using Blue Native Polyacrylamide Gel Electrophoresis (BN-PAGE). *Curr Protoc Mouse Biol* 6, 1-14.
- Sala, D., and Sacco, A. (2016). Signal transducer and activator of transcription 3 signaling as a potential target to treat muscle wasting diseases. *Curr Opin Clin Nutr Metab Care* 19, 171-176.

Figure R2. Stat3 mRNA levels in MPs and FAPs during early stages of skeletal muscle regeneration. Stat3 mRNA abundance per cell in freshly isolated MPs and FAPs from uninjured, 3dpi, and 7dpi 3-month-old male mice (n=3 animals).